# FMRP(1–297)-tat restores ion channel and synaptic function in a model of Fragile X syndrome

Xiaoqin Zhan[1], Hadhimulya Asmara[1], Ning Cheng [2], Giriraj Sahu[1], Eduardo Sanchez[1], Fang-Xiong Zhang[1], Gerald W. Zamponi [1,2], Jong M. Rho[1,2] & Ray W. Turner [1✉]

Fragile X Syndrome results from a loss of Fragile X Mental Retardation Protein (FMRP). We now show that FMRP is a member of a Cav3-Kv4 ion channel complex that is known to regulate A-type potassium current in cerebellar granule cells to produce mossy fiber LTP. Mossy fiber LTP is absent in *Fmr1* knockout (KO) mice but is restored by FMRP(1-297)-*tat* peptide. This peptide further rapidly permeates the blood-brain barrier to enter cells across the cerebellar-cortical axis that restores the balance of protein translation for at least 24 h and transiently reduces elevated levels of activity of adult *Fmr1* KO mice in the Open Field Test. These data reveal that FMRP(1-297)-*tat* can improve function from the levels of protein translation to synaptic efficacy and behaviour in a model of Fragile X syndrome, identifying a potential therapeutic strategy for this genetic disorder.

[1] Hotchkiss Brain Institute, University of Calgary, Calgary, AB T2N 4N1, Canada. [2] Alberta Children's Hospital Research Institute, University of Calgary, Calgary, AB T2N 4N1, Canada. ✉email: rwturner@ucalgary.ca

The loss of Fragile X mental retardation protein (FMRP) in Fragile X syndrome (FXS) can interfere with the translation of numerous proteins important for the normal development of synaptic transmission. A growing body of evidence suggests that behavioral disorders in FXS can involve a disruption in the processing of sensory information in both cortex and cerebellum[1–3]. The mossy fiber projection to cerebellar granule cells represents a major synaptic portal to the cerebellar cortex where long-term plasticity can shape sensory input[4–7]. Any dysfunction at this synapse will thus impair signal processing by the cerebellar cortex at the first stage of sensory input.

Cerebellar granule cells express a Cav3–Kv4 ion channel complex that contributes to long-term potentiation (LTP) of mossy fiber input by increasing the intrinsic excitability of granule cells in the vermal region of lobule 9[8,9]. FMRP is widely expressed in cerebellum and can modulate the activity of ion channels in the membrane, including Kv4 potassium channels[10–14]. Given the central role of Kv4 channels in mossy fiber LTP we investigated the potential for FMRP to shape long-term changes in granule cell excitability.

The present study shows that FMRP is a constituent member of the Cav3–Kv4 complex that modulates both Cav3 and Kv4 channels to reduce Kv4 current amplitude. LTP at the mossy fiber-granule cell input is absent in Fmr1 knockout (KO) mice but rescued by infusing an N-terminal fragment of FMRP (FMRP(1–297)) into granule cells. Moreover, a FMRP(1–297)-tat peptide introduced to Fmr1 KO mice by tail vein injection restores Cav3–Kv4 complex function and mossy fiber LTP, reduces the level of activity in adult animals within 1 h, and rescues disrupted translation of select proteins associated with FXS for at least 24 h, supporting the potential for a tat-FMRP conjugate approach to be developed as a therapeutic agent for FXS.

## Results

**FMRP restores mossy fiber LTP in Fmr1 KO mice.** The reduction in A-type current in granule cells following a theta burst stimulus (TBS) to mossy fibers was traced to a hyperpolarizing shift in the half voltage for Kv4 channel inactivation (Vh) (referred to here as a left-shift in Kv4 Vh)[8]. To determine the potential role for FMRP in regulating Kv4 channels and LTP in granule cells, whole-cell recordings were obtained in the vermis region of lobule 9 from male P16–P22 wild-type (WT) mice or Fmr1 KO mice and mossy fibers were stimulated to evoke a just threshold excitatory postsynaptic potential (EPSP) (Fig. 1a). In 6/6 cells of WT mice a TBS was followed by an initial peak increase in EPSP amplitude that then decreased to an elevated level of $138.8 \pm 11.0\%$ ($n = 6$; $p = 0.012$) of control 5–13 min post TBS (Fig. 1a). In WT cells just subthreshold synaptic stimulation was associated with little firing probability, but TBS invoked an increase in firing probability to single pulse synaptic stimuli that persisted for at least 10 min post stimulation (Fig. 1a)[8,15].

Recordings from granule cells in Fmr1 KO mice revealed similar resting membrane potentials, input resistance and firing threshold as WT mice (Supplementary Table 1). Thus, the loss of FMRP in Fmr1 KO animals did not noticeably influence the basic properties of membrane excitability in granule cells. Yet, in contrast to WT animals, delivering a TBS stimulus to mossy fibers in Fmr1 KO mice failed to evoke LTP of either EPSP amplitude or spike firing probability (Fig. 1b). Previous work has shown that an N-terminal fragment of FMRP (FMRP(1–297)) can modulate select potassium channels[11,16–18]. To test if FMRP(1–297) could restore plasticity at the mossy fiber-granule cell synapse we included 3 nM FMRP(1–297) in the recording electrode. After 10 min equilibration of FMRP(1–297) EPSP amplitude exhibited no significant difference from control 10–15 min post TBS ($103.6 \pm$

10.3%, $n = 7$, $p = 0.74$) (Fig. 1c). However, TBS now evoked a substantial increase in the firing probability that persisted 10–15 min post TBS (Fig. 1c).

We then tested for any effects of FMRP(1–297) on isolated Kv4 current (see "Methods")[8]. In these recordings postsynaptic calcium currents were not blocked, allowing calcium entering via Cav3 channels to interact with the Kv4–KChIP3 complex[8,9,19]. In WT mice mossy fiber TBS evoked a significant left-shift in Kv4 Vh or the half voltage for Kv4 activation (Va) (Fig. 1d). The net effect of these changes on Vh and Va was to produce a $44.2 \pm 11.1\%$ ($n = 7$, $p = 0.011$) reduction in Kv4 current post TBS, indicating a more prominent role of the left-shift in Kv4 Vh on current amplitude (Fig. 1d). Repeating these tests in Fmr1 KO mice revealed no significant difference in the resting values for Vh or Va compared to WT mice (Fig. 1e). However, TBS failed to evoke a left-shift in either Kv4 Vh or Va, and no change in Kv4 current amplitude in Fmr1 KO mice (107 ± 7% of control, $p = 0.40$) (Fig. 1e).

We then tested if FMRP(1–297) could restore the TBS-induced left-shift in Kv4 properties associated with LTP. Including 3 nM FMRP(1–297) in the electrode (Fig. 1f) did not affect the initial control Vh or Va compared to either WT mice (cf Fig. 1d) (Vh, $p = 0.51$; Va, $p = 0.77$) or Fmr1 KO mice recorded with normal electrolyte (Vh, $p = 0.57$; Va, $p = 0.95$) (Fig. 1e). However, TBS in Fmr1 KO cells pre-infused with FMRP(1–297) induced a significant left-shift in Kv4 Vh and Va to reduce Kv4 current by $34 \pm 12\%$ ($n = 6$, $p = 0.044$) (Fig. 1f). We found that in tissue slices preincubated for 2 h and recorded in the presence of 20 μM anisomycin to block protein synthesis mossy fiber TBS still evoked a leftward shift in Kv4 Vh and Va in granule cells of either WT mice or Fmr1 KO mice recorded with 3 nM FMRP(1–297) in the electrode, indicating no requirement for protein translation for these effects (Supplementary Fig. 1).

These results indicate that reintroducing FMRP(1–297) restores the capacity of mossy fiber TBS to evoke a left-shift in Kv4 Vh and Va, and a long-term increase in the probability of spike firing that is absent in Fmr1 KO animals.

**FMRP associates with Cav3.1 and Kv4.3 channels.** To define the association between FMRP and each of Cav3 calcium and Kv4 potassium channels we conducted protein biochemical and imaging assays. Tests on homogenates of WT whole brain (P30–P40) revealed co-immunoprecipitation (coIP) of both Cav3.1 and Kv4.3 channels with FMRP using Cav3.1 and Kv4.3 antibodies to immunoprecipitate (IP) and an N-terminus FMRP antibody to immunoblot (Fig. 2a). The reverse coIP could not be completed as the available N-terminus FMRP antibodies did not function well for IP.

We also tested for fluorescence resonance energy transfer (FRET) by preparing constructs with a fluorescent tag on the N-termini of Cav3.1 and Kv4.3, and on the C-terminus of FMRP or the C-terminal end of FMRP(1–297). Coexpressing a GFP-Cav3.1 construct with FMRP-mKate as donor–acceptor pairs in tsA-201 cells revealed FRET upon excitation at 457 nm (Fig. 2b, c). Similarly, coexpressing GFP-Kv4.3 and FMRP-mKate evoked FRET with 457 nm activation (Fig. 2d, e). Moreover, companion tests confirmed FRET when either GFP-Cav3.1 or GFP-Kv4.3 was coexpressed with a FMRP(1–297)-mKate construct (Supplementary Fig. 2). The specificity of FRET was confirmed by a lack of FRET in cells that expressed either GFP-Cav3.1 or GFP-Kv4.3 together with mKate alone or cells that coexpressed GFP alone with FMRP-mKate or FMRP(1–297)-mKate (Supplementary Fig. 3). These data reveal that FMRP or FMRP(1–297) can associate with Cav3.1 or Kv4.3 with a proximity of less than 10 nm separation when coexpressed in tsA-201 cells, a result

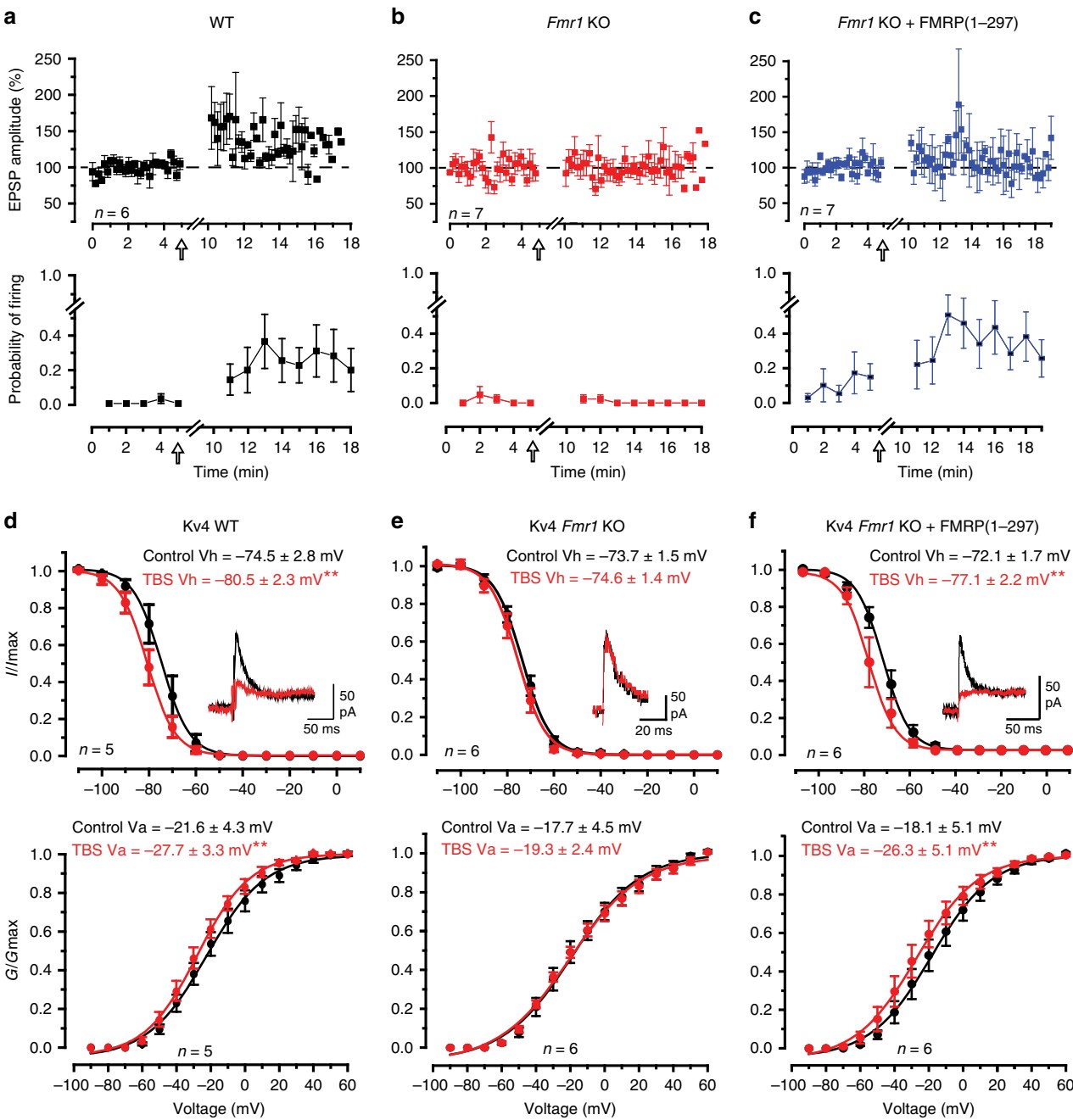

**Fig. 1 Mossy fiber evoked LTP and modulation of Kv4 properties are lost in *Fmr1* KO mice. a–c** Plots of the mean amplitude of the mossy fiber-evoked EPSP and probability of firing per stimulus in whole-cell recordings of lobule 9 granule cells. EPSP amplitudes were only calculated for stimuli that were subthreshold to spike discharge and probability of spike firing was averaged for every 1-min interval (6 stimuli). **a, b** Theta burst stimulation (TBS, indicated by arrow) of mossy fiber input evokes LTP of the EPSP and an increase in probability of firing in granule cells of WT mice (**a**) (% change of EPSP: 138.8 ± 11.0%; firing probability: resting condition 0.5 ± 0.5%, after TBS 25.0 ± 11.4%, *n* = 6 cells from 6 mice) but not *Fmr1* KO mice (**b**) (% change of EPSP: 100.6 ± 5.2%; firing probability: resting condition 1.4 ± 1.5%, after TBS 0.5 ± 0.6%, *n* = 7 cells from 7 mice). **c** Infusing 3 nM FMRP(1–297) into granule cells of *Fmr1* KO mice rescues LTP of spike firing probability but not EPSP amplitude (% change of EPSP: 103.6 ± 10.3%; firing probability: resting condition 9.5 ± 7.5%, after TBS 35.8 ± 10.4%, *n* = 7 cells from 6 mice). **d–f** Plots of the voltage for inactivation and activation of Kv4 current in granule cells in resting conditions (control) and following TBS of mossy fiber input. Insets in (**d–f**) superimpose Kv4 current evoked by a step from −70 to −30 mV for either condition. Following TBS Kv4 Vh and Va are left-shifted in WT mice (**d**) (Vh, *p* = 0.001; Va, *p* = 0.004) but not in *Fmr1* KO mice (**e**) (Vh, *p* = 0.116; Va, *p* = 0.57). **f** Infusing 3 nM FMRP(1–297) into granule cells of *Fmr1* KO mice restores the ability for TBS stimulation to left-shift Kv4 Vh and Va to reduce Kv4 current amplitude within 10 min of introduction (Vh, *p* = 0.004; Va, *p* = 0.008). All mice were P16–P22. Average values are mean ± s.e.m. All statistics were conducted with paired-sample Student's *t* test. **p* < 0.05, ***p* < 0.01. Source data provided as a Source Data file.

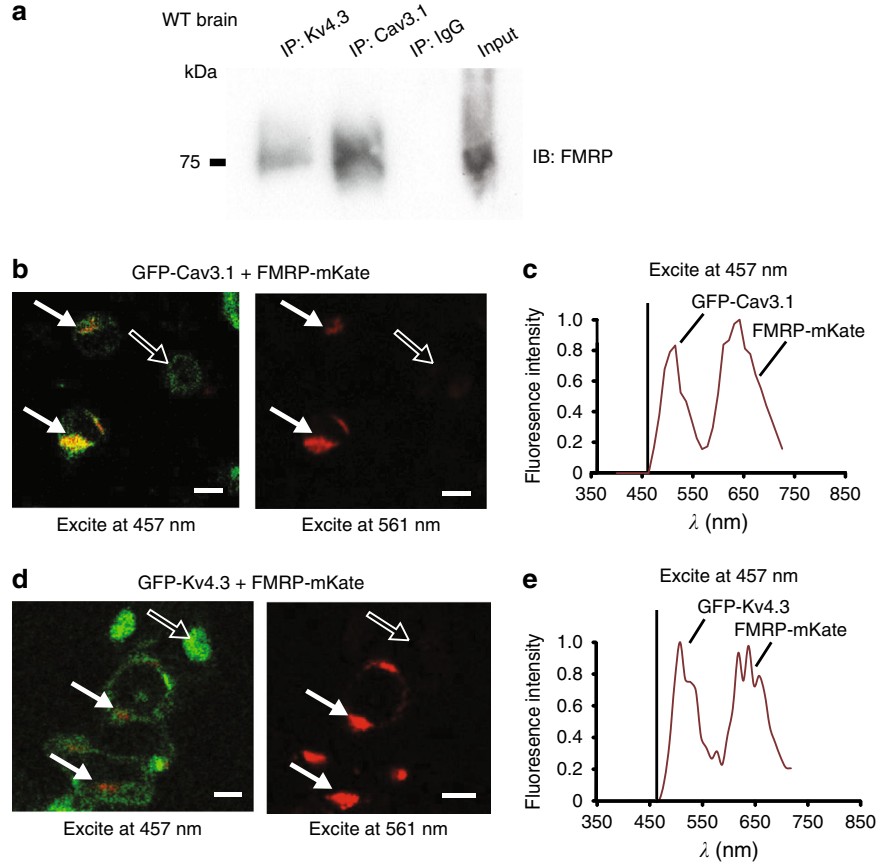

**Fig. 2 Cav3.1 and Kv4.3 channels co-associate with FMRP. a** Co-immunoprecipitation (coIP) of FMRP with Cav3.1 or Kv4.3 from whole brain lysates of WT mice. Immunoprecipitation (IP) was conducted using rabbit anti-Cav3.1 and rabbit anti-Kv4.3 antibodies and rabbit IgG (as control), and immunoblotting using a mouse anti-FMRP N-terminus antibody. All coIP experiments were derived from at least three different experiments. **b**, **c** tsA-201 cells coexpressing GFP-Cav3.1 and FMRP-mKate as a donor–acceptor pair exhibit FRET upon excitation of GFP at 457 nm (**b** solid arrows). A representative example of peak fluorescent emissions for both GFP and mKate upon excitation at 457 nm is shown in (**c**). No dual emissions are detected in cells expressing only GFP-Cav3.1 (open arrows) or in cells expressing both GFP-Cav3.1 and FMRP-mKate when excited at 561 nm (solid arrows) (**b**). **d**, **e** tsA-201 cells coexpressing GFP-Kv4.3 and FMRP-mKate exhibit FRET upon excitation of GFP at 457 nm (solid arrows) that is absent in cells expressing only GFP-Cav3.1 (open arrows) (**d**). A representative example of peak fluorescent emissions for both GFP and mKate upon excitation at 457 nm for cells is shown in (**e**). Scale bars in (**b**, **d**), 10 μm. All FRET experiments were derived from three different experiments with at least three different dishes of cells. Source data are provided as a Source Data file.

consistent with the coIP tests for native FMRP with these channels in whole-brain lysates.

**FMRP modulates Cav3.1 and Kv4.3 channels**. To test the effects of FMRP(1–297) on channel function we expressed Cav3.1 or Kv4.3 channels in tsA-201 cells and compared channel properties in normal electrolyte to those when FMRP(1–297) was included or infused into the electrode. We tested Cav3.1 channels expressed in isolation in tsA-201 cells and found that 30 nM FMRP(1–297) significantly left-shifted Vh and Va (Fig. 3a). While the left shift in Cav3.1 Va should increase channel activation, the amplitude of Cav3.1 current was reduced by $34.6 \pm 8.4\%$ ($n = 7$, $p = 0.0060$ with paired-sample $t$ test), indicating a dominant effect on Vh (Fig. 3a). Kv4.3 Vh in tsA-201 cells was also significantly left-shifted in the presence of FMRP(1–297) but Va was not affected (Fig. 3b). The amplitude of Kv4.3 current in tsA-201 cells was also reduced by $39.1 \pm 5.8\%$ ($n = 5$, $p = 0.0026$ with two-sample $t$ test) after FMRP(1–297) infusion (Fig. 3b). These data are important in indicating that FMRP(1–297) can affect the biophysical properties of Cav3.1 and Kv4.3 channels expressed in isolation in tsA-201 cells.

We repeated these tests to determine if FMRP(1–297) can affect the properties of Kv4 current in *Fmr1* KO granule cells.

Including 30 nM FMRP(1–297) in the internal solution also produced a significant left-shift in Kv4 Vh ($p = 0.037$, two-sample Student's $t$ test) and a reduction of Kv4 current amplitude, but no corresponding shift in Va (Control Va, $-24.0 \pm 3.2$ mV, $n = 10$; Test Va, $-22.6 \pm 3.6$ mV, $n = 10$; $p = 0.78$, two-sample Student's $t$ test) (Fig. 3c). The effects of infusing 30 nM FMRP(1–297) through the electrode (Fig. 3c) did not involve a change in channel insertion in the membrane given no significant change in current density (pA/pF) when tested from $-110$ to $+60$ mV (Supplementary Fig. 4).

Previous work established that Kv4 Vh can be left-shifted by any decrease in Cav3 channel calcium conductance[9,20]. The ability for FMRP(1–297) to left shift the Vh of Cav3.1 recorded in tsA-201 cells might then account for the shift in granule cell Kv4 Vh when this protein is infused. We attempted to record Cav3 current in granule cells of WT and *Fmr1* KO mice but its amplitude in mouse cerebellum was too small to reliably measure. Instead we compared Kv4 Vh in granule cells of *Fmr1* KO mice in the presence of TTA-P2 (1 μM) to block Cav3 current to that of a second set of cells recorded with FMRP(1–297) in the electrode. The cells infused with FMRP(1–297) in the presence of TTA-P2 exhibited a significant left-shift in Kv4 Vh from a mean value of $-74$ to $-78$ mV (Fig. 3d). This is important in indicating that

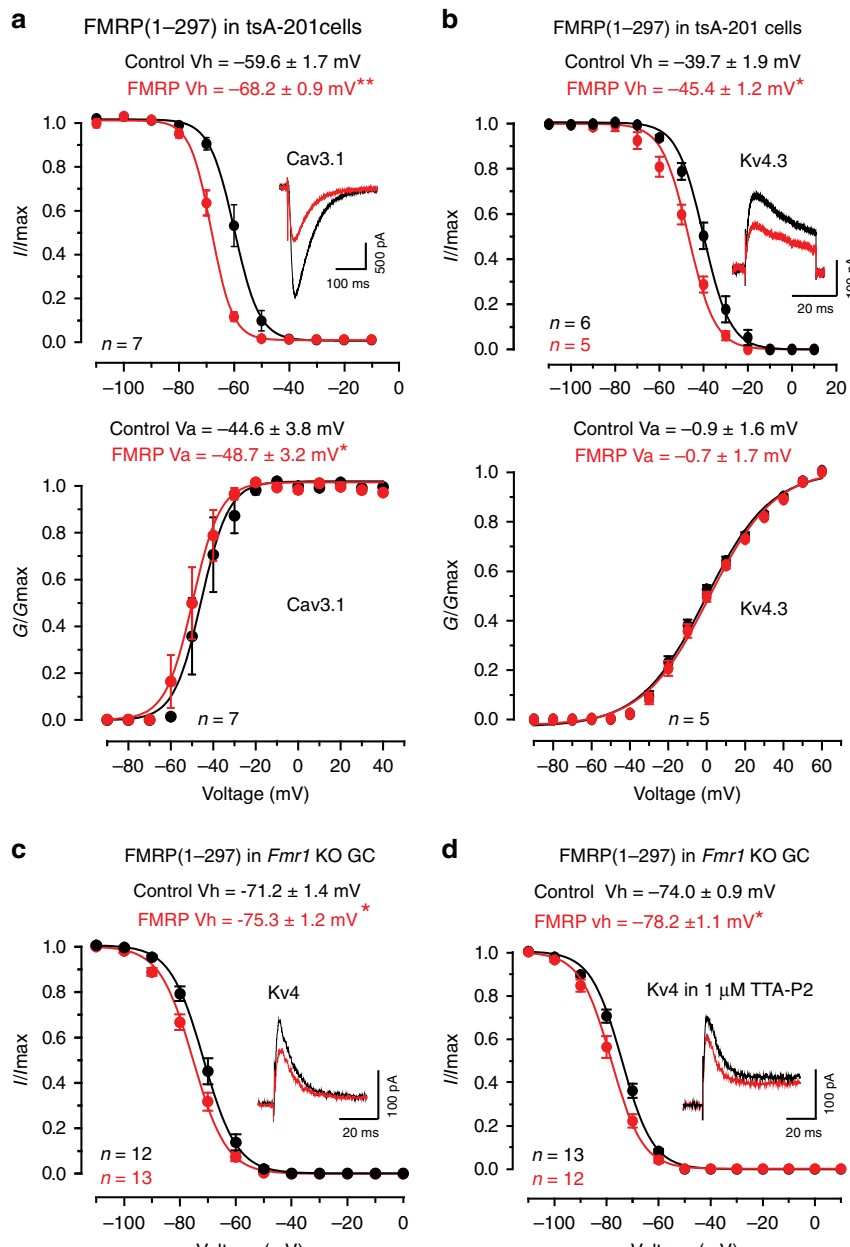

**Fig. 3 FMRP reduces the availability of Cav3.1 and Kv4 current.** FMRP(1–297) (30 nM) was introduced by direct infusion through the electrode (**a**) and in (**b**) using separate cell populations with or without FMRP(1-297) in the electrode. FMRP(1-297) infusion for 10 min promotes a significant left-shift in Vh and Va of Cav3.1 (**a**) (Vh, $p = 0.0021$; Va, $p = 0.02$, both with paired-sample $t$ test), and of the Vh of Kv4.3 (**b**) (Vh, $p = 0.039$, two-sample $t$ test; Va, $p = 0.346$, paired-sample $t$ test) when included in the electrode. Insets show superimposed recordings of current evoked by depolarizing to −30 from −70 mV (Cav3.1) (**a**) or −50 mV (Kv4.3) (**b**). **c** In granule cells of *Fmr1* KO mice including 30 nM FMRP(1-297) in the electrode significantly left-shifts Kv4 Vh ($p = 0.037$ with two-sample $t$ test) and reduces Kv4 current (inset). No effect was detected on Kv4 Va under these conditions. **d** Including 30 nM FMRP(1-297) in the electrode for *Fmr1* KO granule cells induces a left-shift in Kv4 Vh in the presence of 1 μM TTA-P2 to block Cav3-mediated calcium influx ($p = 0.008$ with two-sample $t$ test). Average values are mean ± s.e.m. and sample values at the base of plots in (**c**, **d**) indicate the number of cells drawn from 3 animals. $^*p < 0.05$, $^{**}p < 0.01$. Animals used in (**c**, **d**) were P16–22. GC granule cell. Source data are provided as a Source Data file.

FMRP(1–297) can exert at least some of its effects independently of Cav3.1 calcium conductance on one or more subunits of the Cav3–Kv4 complex.

Together these data indicate that FMRP(1–297) can form very close associations with Cav3.1 and Kv4.3 channel subunits both in tsA-201 cells and in situ, with significant biophysical effects by reducing their availability near resting potential. Despite the rapid effects of FMRP(1–297) infusion on Cav3.1 and Kv4.3 channel Vh (within 10 min) we can not yet attribute these actions to direct interactions between FMRP(1–297) and the channel subunits, as

additional accessory proteins such as KChIP3 and DPP that associate with the Kv4 complex could also be involved[19].

**FMRP(1–297)-*tat* rapidly crosses the BBB**. The fact that FXS stems from the loss of a single protein raises the possibility that reintroducing FMRP would offset the primary factor underlying this genetic disorder. One strategy would be to use a cell penetrating peptide to facilitate passage across the blood–brain barrier (BBB) and cell membranes. For this we prepared a *tat* peptide conjugated to FMRP(1–297) that also included an HA tag for

immunocytochemical tests (Fig. 4a). The final FMRP(1–297)-tat protein was ~35–37 kDa, a size that is within the range for high efficiency transport across the BBB[21]. We validated this construct by testing its effects on whole-cell recordings of Kv4 Vh in granule cells of *Fmr1* KO mice, and found that bath application of even 10 nM FMRP(1–297)-tat significantly left-shifted Kv4 Vh (*Fmr1* KO control Kv4 Vh, $-71.2 \pm 1.9$ mV, $n = 5$; with 10 nM FMRP(1–297)-tat, $-75.6 \pm 2.3$ mV, $n = 5$; $p = 0.0072$; paired sample Student's $t$ test).

We then examined the pattern of FMRP(1–297)-tat protein distribution after its introduction by tail vein injection. FMRP normally exhibits a widespread expression across the brain with high levels detected in cell body layers of the cerebellum, cortex, and hippocampus[12,13,22]. Immunocytochemical tests using an antibody against the FMRP C-terminus confirmed this expression pattern in WT mice on the FVB.129P2 background. In the cerebellum FMRP immunolabel was apparent across all ten lobules with labeling of the granule cell layer in WT mice (Fig. 4b)[12,13]. At higher magnification FMRP immunolabel was detected in virtually all neuronal cell types including granule and Purkinje cells, with label restricted primarily to the cell body regions (Fig. 4c, d). *Fmr1* KO mice injected with vehicle and prepared for immunocytochemistry 30 min later produced no labeling when tested with an antibody against HA (Fig. 4e). We next tail vein injected 1.0 mg/kg HA-FMRP(1–297)-tat in P25–P40 *Fmr1* KO mice (nominally 500 nM plasma concentration at the time of injection) and 30 min or 12 h later processed brains for immunocytochemistry. HA immunolabel was apparent in all layers of the cerebellum within 30 min of injecting HA-FMRP(1–297)-tat, indicating a rapid permeation of the BBB and cerebellar cells (Fig. 4f, g). HA immunolabel was detected within granule and Purkinje cell bodies as well as additional labeling in the molecular layer that did not clearly correspond to Purkinje cell dendrites (Fig. 4f, g). In WT mice an antibody against the FMRP C-terminus detected prominent labeling in the cell body regions of cortical layers, hippocampus, and dentate gyrus, with no clear dendritic label (Fig. 4h). Tail vein injections of HA-FMRP(1–297)-tat into *Fmr1* KO mice again labeled primarily cell bodies of pyramidal cells in the cortex (Fig. 4i–k) and hippocampus (Fig. 4l–n) with exclusion of the nucleus and no apparent labeling of apical or basal dendrites counter labeled with MAP2. As found in cerebellum additional labeling for HA could be found outside neocortical or hippocampal pyramidal cell somas in regions of synaptic inputs (Fig. 4j, m).

In an additional set of tests we tail vein-injected 1.0 mg/kg HA-FMRP(1–297)-tat and confirmed the presence of detectable HA immunolabel in whole-brain sections reacted after fixation at 2, 12, 24, and 48 h and processed/imaged under identical conditions to make qualitative comparisons of HA labeling intensities (Fig. 5). HA immunolabel could be detected at 2 h in cerebellum and with even higher fluorescence intensity in neocortical tissue sections, but not in animals injected with vehicle alone (Fig. 5a, b). By 12 h HA fluorescence was higher in both the cerebellum and neocortex in being detected as a label in cerebellar Purkinje cells and with additional localization in the neuropil and many cell bodies of neocortical regions. By 24 h HA immunofluorescence intensities had dropped in both locations and by 48 h immunofluorescence for HA immunolabel was just detectable (Fig. 5a, b). An interaction between HA-FMRP(1–297)-tat and Cav3.1 and Kv4.3 channels in *Fmr1* KO mice was further verified by obtaining a coIP between HA and these channels 5 h after tail vein injection ($n = 3$) (Fig. 5c).

**FMRP(1–297)-tat is nontoxic to cultured granule cells.** To test for potential toxicity of the FMRP(1–297)-tat peptide we prepared dissociated cultures of cerebellar granule cells from *Fmr1* KO mice and at 3 days in vitro (DIV) applied HA-FMRP (1–297)-tat over a range of 1–500 nM (Fig. 6b, c). Dual labeling for MAP2 and HA confirmed that uptake of FMRP(1–297)-tat by cultured granule cells occurred within 2 h of exposure (Fig. 6a–c). Cells were exposed to treatments for either 24 h or 5 days (culture medium changed every 2 days) before flow cytometric analysis of labeling using a live-dead cell kit. Exposure to either vehicle alone or up to 500 nM FMRP(1–297)-tat showed no significant increase in the number of compromised cells up to 5 days following the treatment (Fig. 6d–g; Supplementary Fig. 5).

These results reveal that FMRP(1–297)-tat peptide can cross the BBB and gain substantial access to cells across the cerebellum and other brain regions within 30 min of injection. Moreover, the FMRP(1–297)-tat peptide did not compromise cell health over time even when applied directly to cultured granule cells at a dose of 500 nM.

**FMRP(1–297)-tat restores mossy fiber-granule cell LTP.** To test the effects of FMRP(1–297)-tat on cell activity we tail vein injected FMRP(1–297)-tat peptide into P16–P22 *Fmr1* KO mice and 2 h later prepared in vitro slices to test the ability to evoke LTP. Injecting 1.0 mg/kg FMRP(1–297)-tat promoted LTP of the EPSP amplitude 10–15 min post TBS and reliably increased spike firing probability for at least 15 min following TBS (Fig. 7a, b). Indeed, the degree of EPSP potentiation and increase in firing probability was strikingly similar to that detected in WT mice (cf. Figs. 1a and 7a), indicating that systemic administration of FMRP (1–297)-tat promoted greater EPSP potentiation than did direct postsynaptic infusion of FMRP(1–297). In contrast, injecting 0.2 mg/kg FMRP(1–297)-tat did not rescue mossy fiber LTP (Fig. 7c, d), even though FMRP immunolabel can be detected in cerebellum 2 h after tail vein injection (Supplementary Fig. 6). Similarly, tail vein injection of *tat* epitope alone (1.0 mg/kg) showed no LTP in response to mossy fiber TBS when slices were prepared 2 h later (Fig. 7e, f).

These data functionally validate the ability for FMRP(1–297)-tat to access cerebellar neurons in vivo and reveal a concentration-dependent effect on its ability to restore synaptic plasticity. The finding that 1.0 mg/kg FMRP(1–297)-tat injection promoted LTP that was comparable to WT mice is also important in suggesting an additional potential influence of FMRP(1–297)-tat on presynaptic elements not accessed by postsynaptic infusion of FMRP(1–297).

**FMRP(1–297)-tat reduces elevated activity in *Fmr1* KO mice.** Hyperactivity and anxiety are symptoms exhibited by FXS patients[23,24] that have been assessed in the *Fmr1* KO mouse model in terms of activity levels in the open field test (OFT)[25–27]. We therefore used OFT to quantify the extent to which tail vein injection of FMRP(1–297)-tat might rescue elevated levels of activity in *Fmr1* KO mice. For this we tail vein injected 1.0 mg/kg FMRP(1–297)-tat in male P55–P100 mice. In agreement with previous work, vehicle-treated *Fmr1* KO mice traveled a significantly greater distance and with a higher velocity during a 30 min period than vehicle-treated WT mice (Fig. 8). We then injected WT and *Fmr1* KO animals with different amounts of FMRP(1–297)-tat and conducted OFT 1, 24, or 48 h later. A two-way ANOVA analysis indicated a genotype-dependent effect of FMRP(1–297)-tat (interaction of genotype and treatment, $p = 0.012$). One hour after administering 1.0 mg/kg FMRP(1–297)-tat, *Fmr1* KO mice traveled significantly less than vehicle injected *Fmr1* KO animals, while WT mice injected with 1.0 mg/kg FMRP(1–297)-tat showed no significant change (Fig. 8a, left panel). One-way ANOVA revealed a concentration-dependent effect of FMRP

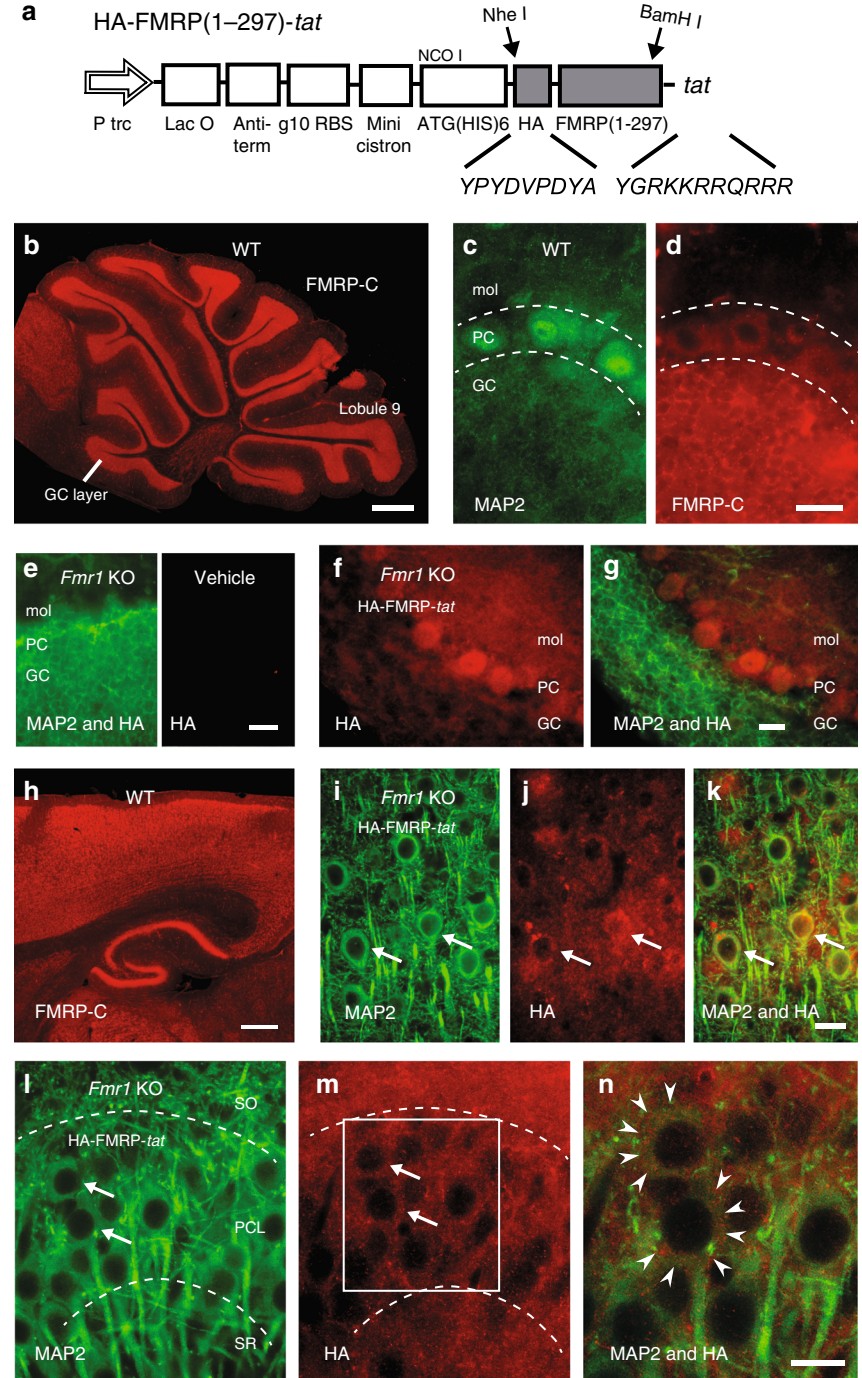

**Fig. 4 A FMRP(1–297)-*tat* construct rapidly crosses the BBB to enter central neurons. a** Schematic diagram of HA-FMRP(1–297)-*tat* construct. HA sequence and 11 aa tat sequence were inserted through the two restriction enzyme sites indicated by *arrows*. **b–d** Immunolabel distribution for an FMRP C-terminus (FMRP-C) antibody in WT mouse cerebellum with expanded views of dual labeled MAP2 (**c**) and FMRP-C (**d**) in lobule 9 cell layers. **e–g** Cerebellar sections from *Fmr1* KO mice that were tail vein injected with either vehicle (**e**) or 1.0 mg/kg HA-FMRP(1–297)-*tat* (**f**, **g**) and processed 30 min later for MAP2 and HA labeling. HA label is absent in vehicle-injected *Fmr1* KO mouse cerebellum (**e**) but is detected in all cell layers after tail vein injection of HA-FMRP(1–297)-*tat* (**f**, **g**). **h** Low power view of immunolabel for the FMRP-C antibody in WT mouse in the region of neocortex and hippocampus. **i–n** Magnified views of *Fmr1* KO mouse neocortex at 30 min (**i–k**) and CA3 hippocampus at 12 h (**l–n**) following tail vein injection of 1.0 mg/kg HA-FMRP(1–297)-*tat* and dual labeled for MAP2 and HA. Boxed region in (**m**) is magnified in (**n**), with the boundaries of cell membranes of hippocampal pyramidal cells denoted by *arrowheads* in (**n**) to distinguish FMRP(1–297) immunolabel in cytoplasmic regions. Dashed lines in (**c**, **d**) and (**l**, **m**) delineate major cell boundaries and arrows in (**l**, **m**) highlight representative cells exhibiting both MAP2 and HA immunolabels. Animals used for immunolabeling ranged from P25–P40, with immunolabel experiments conducted in at least 3 separate animals in (**b–n**). GC granule cell layer, PC Purkinje cell layer, mol molecular layer, SR stratum radiatum, PCL pyramidal cell layer, SO stratum oriens. Scale bars: **b**, **h**, 500 μm; **c–g**, **i–n**, 20 μm.

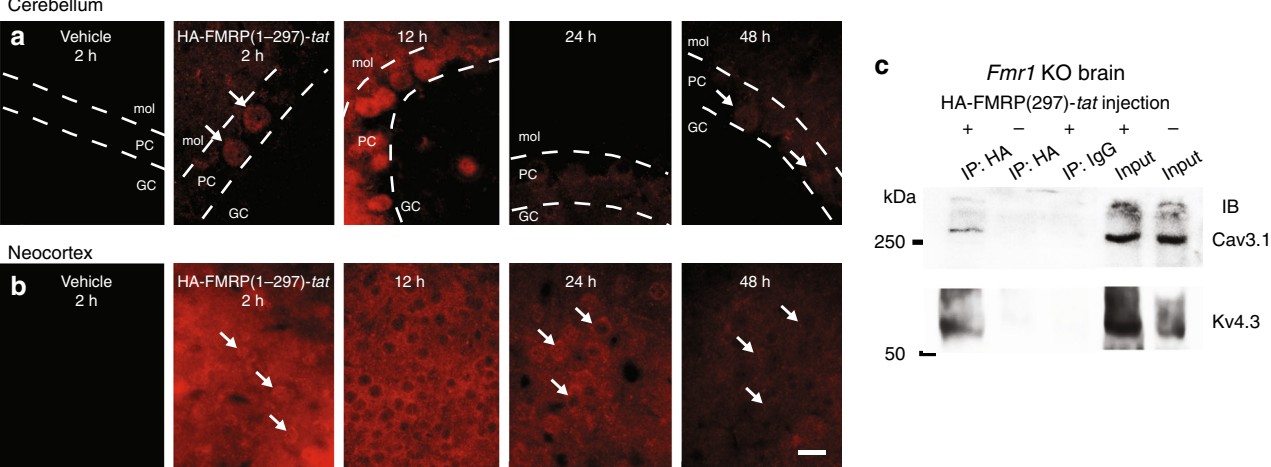

**Fig. 5 HA-FMRP(1–297)-*tat* immunolabel persists at least 24 h in *Fmr1* KO mice. a, b** Qualitative comparison between HA immunolabel intensity in cerebellar tissue sections (**a**) or neocortex (**b**) prepared at the indicated times following tail vein injection of vehicle or 1.0 mg/kg HA-FMRP(1–297)-*tat* into P30–P60 *Fmr1* KO mice. All sections were reacted simultaneously with anti-HA antibody and imaged/processed at identical camera settings and light intensities to enable qualitative comparisons. Arrows identify the position of representative cells positive for HA immunolabel. **c** CoIP of HA-FMRP(1–297)-*tat* with Cav3.1 and Kv4.3 in brain lysates of *Fmr1* KO mice 5 h after tail vein injection of HA-FMRP(1–297)-*tat*. IP was conducted using a mouse anti-HA antibody and mouse normal IgG (as control) and immunoblotting using rabbit anti-Cav3.1 and rabbit anti-Kv4.3 antibodies after an injection of 27 mg/kg HA-FMRP(1–297)-*tat*. All immunolabel data in (**a, b**) and coIP tests in (**c**) were derived from at least three different animals. Dashed lines in (**a**) denote cell layer boundaries. Scale bar for (**a, b**), 20 µm. PC Purkinje cell layer, GC granule cell layer, mol molecular layer. Source data are provided as a Source Data file.

(1–297)-*tat*, with a significant reduction in the distance traveled by *Fmr1* KO mice for 1.0 mg/kg but not for 0.2 or 2.0 mg/kg FMRP (1–297)-*tat* (Fig. 8a, right panel). The same pattern was detected at 1 h post injection for measures of velocity (Fig. 8d). Injecting 1.0 mg/kg *tat* epitope alone to *Fmr1* KO mice had no effect on either distance or velocity measurements (Fig. 8a, d, right panels). Twenty-four hours after injecting FMRP(1–297)-*tat* the distance and velocity were slightly reduced from vehicle-treated *Fmr1* KO mice at all doses, but not to significant levels (Fig. 8b, e). Forty-eight hours after FMRP(1–297)-*tat* injections the levels of activity had returned to levels not significantly different from vehicle-treated *Fmr1* KO mice (Fig. 8c, f).

**FMRP(1–297)-*tat* restores protein levels in *Fmr1* KO mice.** A second established role for FMRP is regulation of protein translation, where loss of FMRP alters the expression levels of multiple proteins that contribute to circuit dysfunction[14,28–32]. To determine if FMRP(1–297)-*tat* could influence protein translation we used Western blots in lysates of cerebellum or brain to measure the levels of a select set of proteins previously shown to be altered in *Fmr1* KO mice. These tests established that αCaMKII and the amyloid precursor protein (APP) are significantly elevated in cerebellar and brain lysates of P21–30 *Fmr1* KO compared to WT mice (Fig. 9a, b). In contrast, the level of PSD95 protein was lower in *Fmr1* KO cerebellar (but not brain) lysates compared to WT (Fig. 9c). A single tail vein injection of 1.0 mg/kg FMRP(1–297)-*tat* significantly reduced these differences in protein levels in both cerebellum and brain samples even 24 h after injection, without affecting the similar baseline values for PSD95 in brain samples. Interestingly, the levels of all three proteins in WT animals did not significantly change after FMRP(1–297)-*tat* injections (Fig. 9).

## Discussion

Several studies have attempted to restore FMRP in *Fmr1* KO mice through viral injections, *tat*-conjugated peptides, or by targeting the hypermethylation that blocks transcription of the *Fmr1* gene[33–37]. Success was gained by reintroducing FMRP by viral

transfection but this was offset by variability in the distribution and expression levels of FMRP, and deleterious effects if over-expressed[35,37,38]. Others have used a pharmacological approach to counteract downstream signaling molecules that are disrupted when FMRP is lost[39,40]. Another strategy is to use a *tat*-conjugate peptide to increase access across the BBB and cell membranes. The only other study that explored a *tat*-conjugate approach tested a full-length FMRP[36] and reported toxicity on cultured fibroblasts by ~7.5 nM, dampening enthusiasm for a *tat*-conjugate approach to treating FXS. We tested an N-terminal fragment of FMRP that has been shown to affect specific ion channels[11,16–18]. The results indicate that FMRP(1–297)-*tat* has the ability to shift the biophysical properties of Cav3.1 and Kv4.3 channels, rescue LTP of mossy fiber input to cerebellum, restore levels of proteins in *Fmr1* KO mice for at least 24 h, and reduce elevated levels of activity in *Fmr1* KO animals at 1 h post-FMRP-*tat* injection, but not at 24 or 48 h post-injection. No toxicity was detected up to 5 days after direct exposure of cultured cells to 500 nM FMRP(1-297)-*tat*. Together these data indicate the ability to use a *tat* peptide-based approach to restore FMRP-related circuit function in the mouse model of FXS.

Our tests using immunocytochemistry indicate that HA-FMRP (1–297)-*tat* is rapidly taken up following tail vein injection and distributed widely across the cerebellar-cortical axis. Most of the structures known to label for FMRP in WT mice[12,13] were labeled after HA-FMRP(1–297)-*tat* injection, with a preference for uptake and/or sequestration at the soma. Additional labeling could be detected outside neuronal somata, such as in layers with a prominent synaptic component (i.e., molecular layer of cerebellum). The extent to which this labeling reflects uptake by glial or synaptic components is unknown. We also can not ensure that all structures will take up FMRP(1–297)-*tat* to the levels required for a desired function, even though mossy fiber LTP was rapidly restored using the concentrations tested here. However, the results are extremely promising in revealing uptake of FMRP (1–297)-*tat* across the brain axis in a very short period of time and retention of the HA immunolabel tag or FMRP(1–297) up to 48 h later.

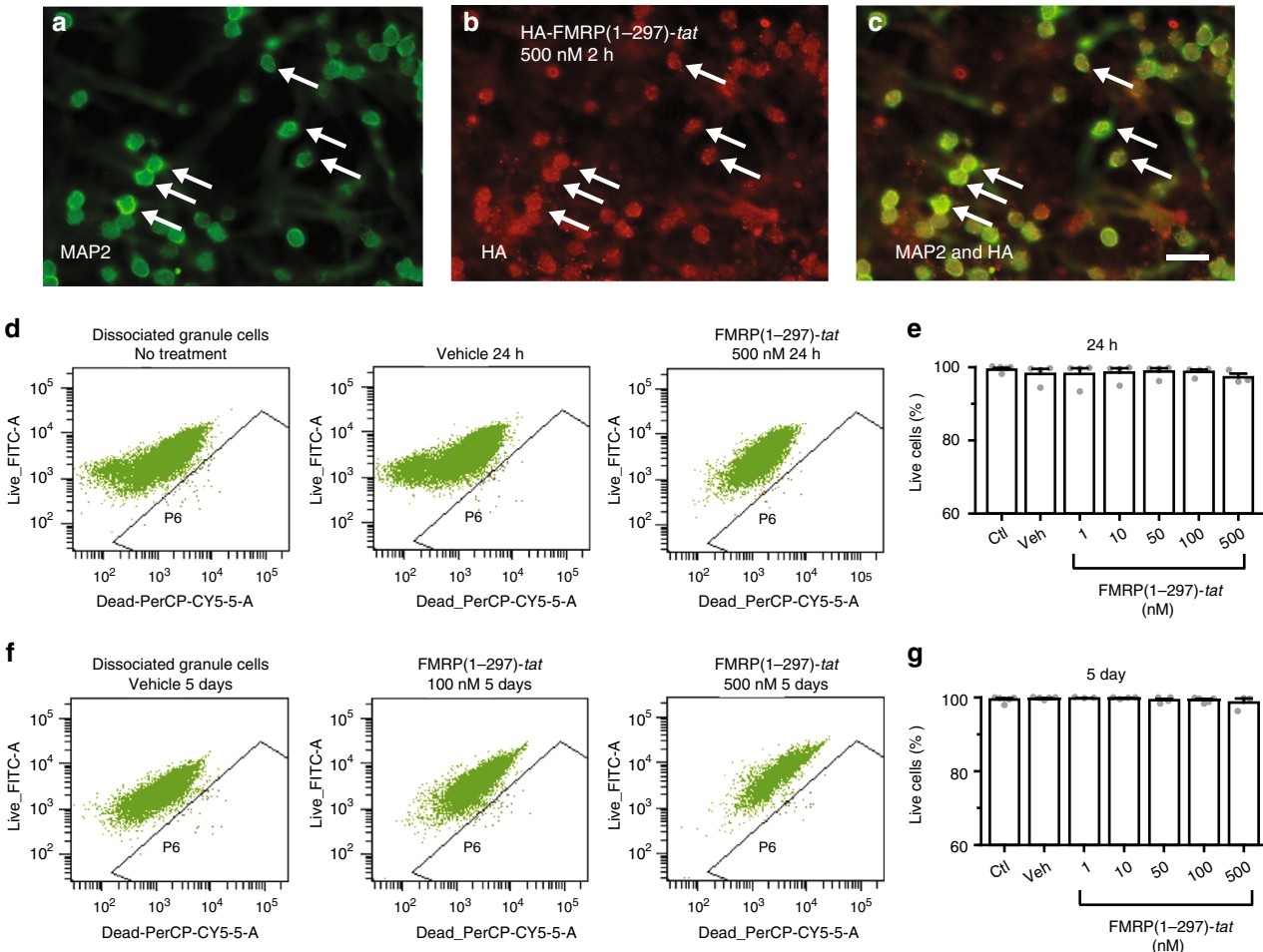

**Fig. 6 FMRP(1–297)-*tat* is non-toxic to cultured *Fmr1* KO granule cells. a–c** Images of dual immunolabeled dissociated cultures of *Fmr1* KO granule cells at 3 DIV for MAP2 and an anti-HA antibody 2 h following direct exposure to 500 nM (equivalent to 1.0 mg/kg in situ) HA-FMRP(1-297)-*tat*. Arrows denote position of representative cells labeling for both MAP2 and HA. Immunostaining was repeated with 3 different batches of cerebellar granule cell cultures. Scale bar, 25 µm. **d–g** Flow cytometric analysis of a live-dead cell viability assay in cerebellar granule cell cultures 24 h (**d**, **e**) and 5 days (**f**, **g**) after single exposure to the indicated concentrations of FMRP(1–297)-*tat*. The percentage of live cells in each cell population is not significantly different among the control group with no treatment, the vehicle treated group and cell groups treated with FMRP(1–297)-*tat* for 24 h (control, 99.4 ± 0.4%, $n = 5$; vehicle, 99.6 ± 0.2%, $n = 5$; 1 nM, 99.7 ± 0.1%, $n = 3$; 10 nM, 99.7 ± 0.1%, $n = 4$; 50 nM, 99.2 ± 0.4%, $n = 4$; 100 nM, 99.3 ± 0.3%, $n = 5$; 500 nM, 98.6 ± 1.1%, $n = 3$; $F(6,28) = 0.763$, $p = 0.607$, one-way ANOVA) and at 5 days (vehicle, 98.2 ± 1.3%, $n = 4$; 1 nM, 98.1 ± 1.6%, $n = 4$; 10 nM, 98.5 ± 1.2%, $n = 4$; 50 nM, 98.8 ± 0.9%, n = 4; 100 nM, 98.7 ± 0.6%, $n = 4$; 500 nM, 97.2 ± 1.0%, $n = 3$; $F(6,27) = 0.406$, $p = 0.867$, one-way ANOVA). *N* value represents the number of independent experiments repeated. Average values are mean ± s.e.m. Ctl Control, Veh vehicle. Source data are provided as a Source Data file.

Tail vein injection of 1.0 mg/kg FMRP(1–297)-*tat* was sufficient to achieve therapeutic actions from the cellular to behavioral level in 1–2 h. We found a progressive increase in HA immunolabel up to 12 h in terms of fluorescence intensity in tissue sections, with just detectable levels by 48 h. However, we do not know if the molecule remains fully intact or is in the process of degradation/elimination, as suggested by a lower signal level at 48 h. It is possible that future work will identify suitable shorter length FMRP constructs or modifications to the *tat* peptide that will further improve the speed of transport across the BBB or the retention of FMRP(1–297) in situ[41–43].

Fragile X patients exhibit symptoms that have been correlated to the degree of hypoplasia of the posterior cerebellar vermis[3,44,45]. The results obtained here on cerebellar synaptic plasticity and cell output are thus relevant to the conditions inherent to patient populations and to lobule 9 granule cells where the work is centered. The actions of FMRP(1–297)-*tat* were detected at multiple levels of cell and circuit function.

Previous work has shown that intracellular infusion of the N-terminal fragment FMRP(1–297) can increase activation of Slack,

SK2, BK, and Kv1.2 potassium channels[11,16–18,46]. The molecular sites for interaction were identified between FMRP(1–297) and C-terminal regions of both Kv1.2[11] and Slack potassium channels[16]. The ability to detect an interaction with Kv1.2 depends on the state of phosphorylation[11], identifying an additional factor to consider when testing FMRP(1–297) interactions. The current study identifies Cav3.1 calcium and Kv4.3 potassium channels as two previously unrecognized interactors with the FMRP N-terminus (1–297), raising the number of channel isoforms that can be modified by this FMRP fragment to six. In the case of Kv4.3 the modulation was apparent at the level of biophysical properties without a change in Kv4.3 channel density. Further work will thus be needed to define how an N-terminal FMRP fragment interacts at the molecular level with each member of a Cav3–Kv4–KChIP3 complex that together regulate Kv4 current amplitude[19,47–49].

FMRP is known to play a key role in regulating protein translation[28,30–32,50]. Here, we found that FMRP(1–297)-*tat* reduced differences in protein levels detected between *Fmr1* KO and WT mice regardless of whether the levels were initially

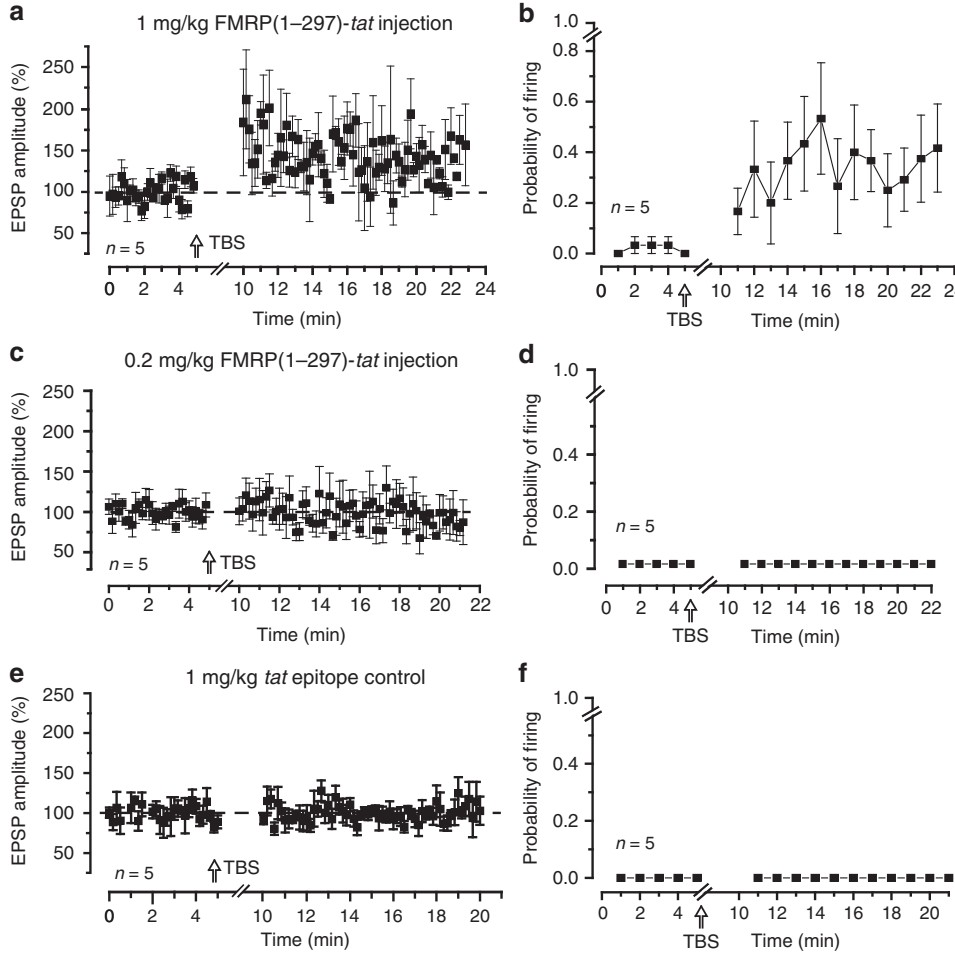

**Fig. 7 FMRP(1–297)-*tat* restores LTP at the mossy fiber-granule cell synapse of *Fmr1* KO mice.** Shown are plots of mossy fiber-evoked EPSP amplitude and spike firing probability measured in granule cells from *Fmr1* KO mice in slices prepared 2 h following tail vein injections. EPSP amplitudes were calculated only for stimuli that were subthreshold to spike discharge. LTP in response to TBS of mossy fibers is restored to near WT levels by prior injection of 1.0 mg/kg FMRP(1–297)-*tat* in terms of EPSP amplitude (**a**) (% change in EPSP: 141.0 ± 23.2%, *n* = 5 cells from 5 mice), firing probability (**b**) (resting condition 2.0 ± 2.2%, after TBS 36.8 ± 18.4%, *n* = 5 cells from 5 mice), but with no effect following 0.2 mg/kg FMRP(1–297)-*tat* injection (**c**, **d**) (% change in EPSP: 99.6 ± 10.9%, *n* = 5 cells from 5 mice) or injection of the tat epitope alone (**e**, **d**) (% change in EPSP: 100.4 ± 1.2%, *n* = 5 cells from 4 mice; no spikes fired in resting condition or after TBS was delivered). Average values are mean ± s.e.m. Source data are provided as a Source Data file.

increased (αCaMKII, APP) or decreased (PSD95) in *Fmr1* KO animals. Moreover, the effects of FMRP(1–297)-*tat* on protein levels were apparent 24 h after a single tail vein injection, indicating actions on cellular function in both the cerebellum and cortex for an even longer timeframe than the behavioral differences detected in the OFT. These results have broad implications given the number of RNA binding sites identified for FMRP that might benefit from a *tat*-conjugate approach to reintroducing FMRP(1–297).

Our data reveal that FMRP is a constituent member of the Cav3-Kv4 complex with a role for FMRP in promoting LTP at the mossy fiber synapse, a function it shares with other central synapses by enabling long-term potentiation or depression[37,51]. Indeed, the short time required for FMRP(1–297) infusion to restore the capacity to evoke LTP suggests that FMRP is a critical element in the pathways activated by mossy fiber stimulation[8]. Tail vein injections of 1.0 mg/kg FMRP(1–297)-*tat* also had a greater influence on EPSP potentiation and granule cell firing rate than direct infusion of FMRP(1–297) into granule cells. The full effects of FMRP(1–297)-*tat* on mossy fiber-evoked LTP must then be exerted on additional elements accessed by the *tat*-conjugate peptide that are not influenced by direct postsynaptic infusion. It is important to clarify that we can not conclude that

the rescue of mossy fiber plasticity by FMRP(1–297)-*tat* is directly linked to a reduction in elevated levels of activity in the OFT. Rather, we focused on the mossy fiber-granule cell synapse as a representative synapse to gauge the effectiveness of introducing FMRP(1–297)-*tat* on ionic and synaptic function. Activity in the OFT can be interpreted in the context of either hyperactivity or anxiety, both of which are characteristic of Fragile X patients[23,24]. Further work will be needed to distinguish between these possibilities. However, measures of activity in the OFT can be assumed to reflect the combined output of multiple brain regions that could be influenced by HA-FMRP-*tat* that distributes widely over the cerebellar-cortical axis and restores protein levels in both cerebellar and whole brain lysates. While the full range of sites affected by HA-FMRP-*tat* remain to be determined, the results are important in reviving the potential to use a *tat* peptide-based approach to replace an active fragment of FMRP to rescue circuit function in FXS.

## Methods

**Mouse lines**. Wild type (Jackson Lab stock #004828, FVB.129P2-*Pde6b*+ *Tyr*<sup>c-ch</sup>/AntJ) and *Fmr1* knockout (Jackson Lab stock #004624,FVB.129P2-*Pde6b*+ *Tyr*<sup>c-ch</sup> *Fmr1*<sup>tm1Cgr</sup>/J) mice on FVB background were purchased from Jackson Lab and maintained in an Animal Resource Center of the University of Calgary in

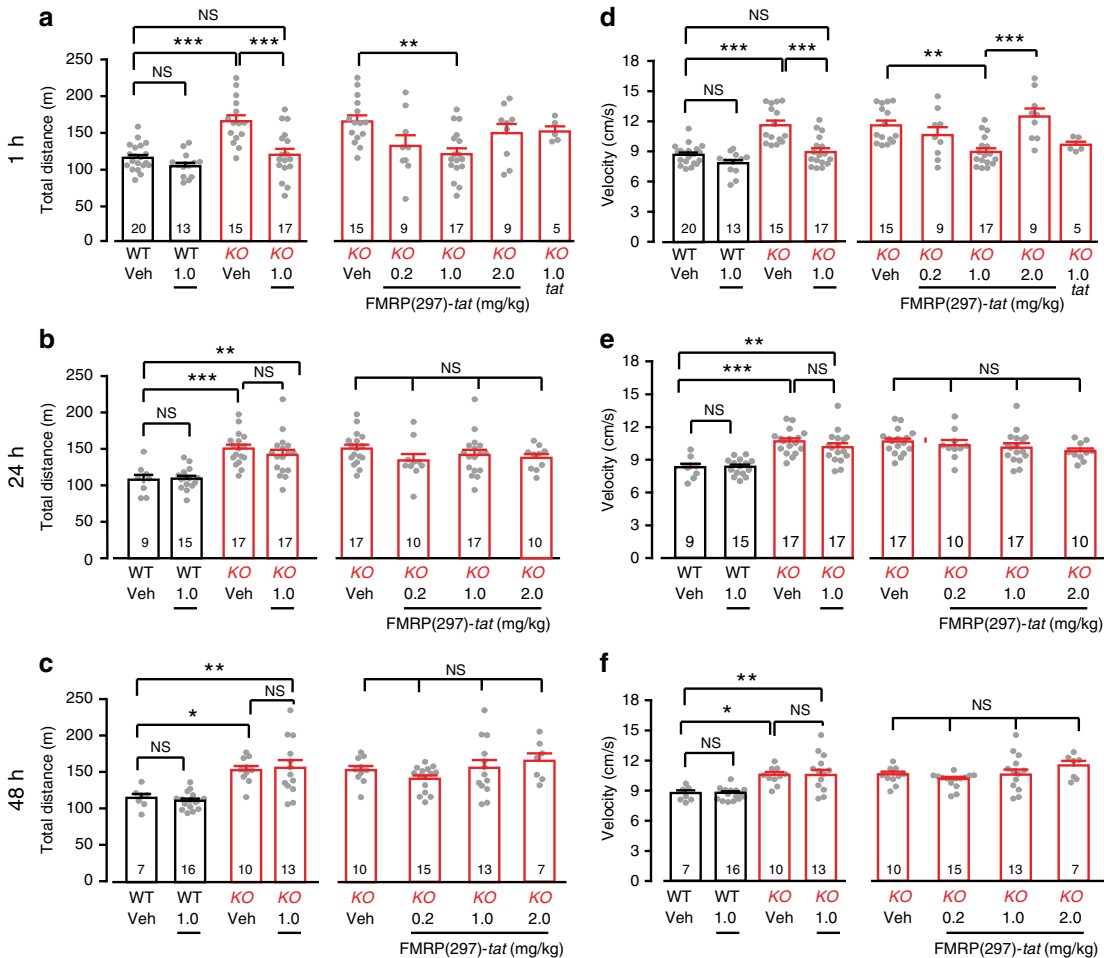

**Fig. 8 FMRP(1–297)-*tat* reduces elevated levels of activity in *Fmr1* KO mice.** Shown are mean bar-dot plots for total distance (**a–c**) and mean velocity (**d–f**) in P55-100 WT and *Fmr1* KO mice over 30 min in an open field test (OFT) at 1, 24, and 48 h following tail vein injection of vehicle, different concentrations of FMRP(1–297)-*tat*, or *tat* epitope alone. *Fmr1* KO mice exhibit higher levels of activity compared to WT animals in terms of total distance traveled in the field that is significantly reduced at 1 h by 1.0 mg/kg FMRP (1-297)-*tat* tail vein injection (**a** left panel: two-way ANOVA analysis of the interaction between genotype and treatment $F_{(1,61),interaction} = 6.792$, $p = 0.012$; post hoc Tukey's test: $p_{(KO\ veh,WT\ veh)} < 0.0001$, $p_{(KO\ veh,KO\ 1.0)} < 0.0001$; **a** right panel: one-way ANOVA $F_{(4,54)} = 3.817$, $p = 0.00875$; post-hoc Tukey's test: $p_{(KO\ veh,KO\ 1.0)} = 0.00475$), as well as the velocity of movement (**d**, left panel: Two-way ANOVA $F_{(1,61),interaction} = 7.035$, $p = 0.0102$; post hoc Tukey's test: $p_{(KO\ veh,WT\ veh)} < 0.0001$, $p_{(KO\ veh,KO\ 1.0)} < 0.0001$; **d** right panel: one-way ANOVA $F_{(4,54)} = 6.943$, $p = 1.6 \times 10^{-4}$; post hoc Tukey's test: $p_{(KO\ veh,KO\ 1.0)} = 0.00187$). By comparison the level of activity in *Fmr1* KO mice is not affected by 1.0 mg/kg *tat* epitope alone (**a** right panel: post hoc Tukey's test $p_{(KO\ veh,KO\ tat)} = 0.937$; **d** right panel: post hoc Tukey's test $p_{(KO\ veh,KO\ tat)} = 0.268$). Similarly, injection of 1.0 mg/kg FMRP(1–297)-*tat* in WT animals had no significant effect on distance or velocity for any of the 1, 24, or 48 h time points (left panels for distance: **a** post hoc Tukey's test: $p_{(WT\ veh,WT\ 1.0)} = 0.629$; **b** $p_{(WT\ veh,WT\ 1.0)} = 0.999$; **c** $p_{(WT\ veh,WT\ 1.0)} = 0.636$; left panels for velocity: **d** $p_{(WT\ veh,WT\ 1.0)} = 0.324$; **e** $p_{(WT\ veh,WT\ 1.0)} = 0.9998$; **f** $p_{(WT\ veh,WT\ 1.0)} > 0.9999$). Animals used for behavioral tests ranged from P55–P100. Average values are mean ± s.e.m, and sample values identified at the base of bar plots represent the number of animals. *$p < 0.05$, **$p < 0.01$, ***$p < 0.001$, NS not significant. Source data are provided as a Source Data file.

accordance with ethical guidelines of the Canadian Council of Animal Care reviewed and approved by a Cumming School of Medicine Animal Care Committee. All animals had free access to water and food with a daily temperature of 20–23 °C and relative humidity of 40–60%. The light cycle was controlled as 12 h light and dark cycle. For electrophysiology and behavioral tests, male mice were used. For Western blot and coimmunoprecipitation tests samples were harvested from both males and females.

**Brain slice preparation**. Sagittal cerebellar sections for electrophysiological recordings were prepared from P16 to P22 male mice as previously described[19]. Briefly, animals were anaesthetized by isoflurane inhalation and the cerebella were dissected out and placed in ice-cold artificial cerebrospinal fluid (aCSF) composed of (in mM): 125 NaCl, 25 NaHCO$_3$, 25 D-glucose, 3.25 KCl, 1.5 CaCl$_2$, 1.5 MgCl$_2$ bubbled with carbogen (95% O$_2$ and 5% CO$_2$) gas. Tissue slices of 260 μm thickness were cut by vibratome (Leica VT1200 S) and allowed to recover for 20–30 min at 37 °C and then at room temperature (RT) (25 °C) in aCSF bubbled with carbogen gas. For recordings slices were moved to a recording chamber which temperature

was maintained at 31–33 °C on the stage of an Olympus BX51W1 microscope. Unless otherwise indicated all chemicals were obtained from Sigma-Aldrich.

**Dissociated granule cell culture and flow cytometry**. Cerebella were dissected out from anesthetized P6 *Fmr1* KO mice, diced and trypsinized in Hanks' balanced salt solution/bovine serum albumin (BSA) dissection solutions[52]. Isolated cells were plated onto 24-well plates coated with poly-D-lysine (1 μg/ml) at a total number of $1 \times 10^6$ each well. Cultured cells were incubated at 37 °C under 5% CO$_2$ in Dulbecco's modified Eagle's medium (DMEM) supplemented with 10% fetal calf serum, insulin (5 μg/ml), KCl (25 mM) and 1% pen–streptomycin. After culture for 24 h, cytosine β-D-arabinofuranoside (5 μM) (Sigma-Aldrich) was added to the culture medium for 24 h to inhibit the proliferation of nonneuronal cells and medium was changed every two days. A single dose of different concentrations of FMRP(1–297)-*tat* peptide was applied in the granule cell culture medium at 3 DIV or 7 DIV. At 8 DIV cells were labeled with a live-dead assay kit (ab115347, Abcam) before being analyzed with a flow cytometer BD LSR II (BD Biosciences) at 488 nm excitation wavelength. Emission of labeled dye was detected at 530 nm for FITC channel and 695 nm for PerCP-CY5-5 channel. Forward scatter, side scatter and

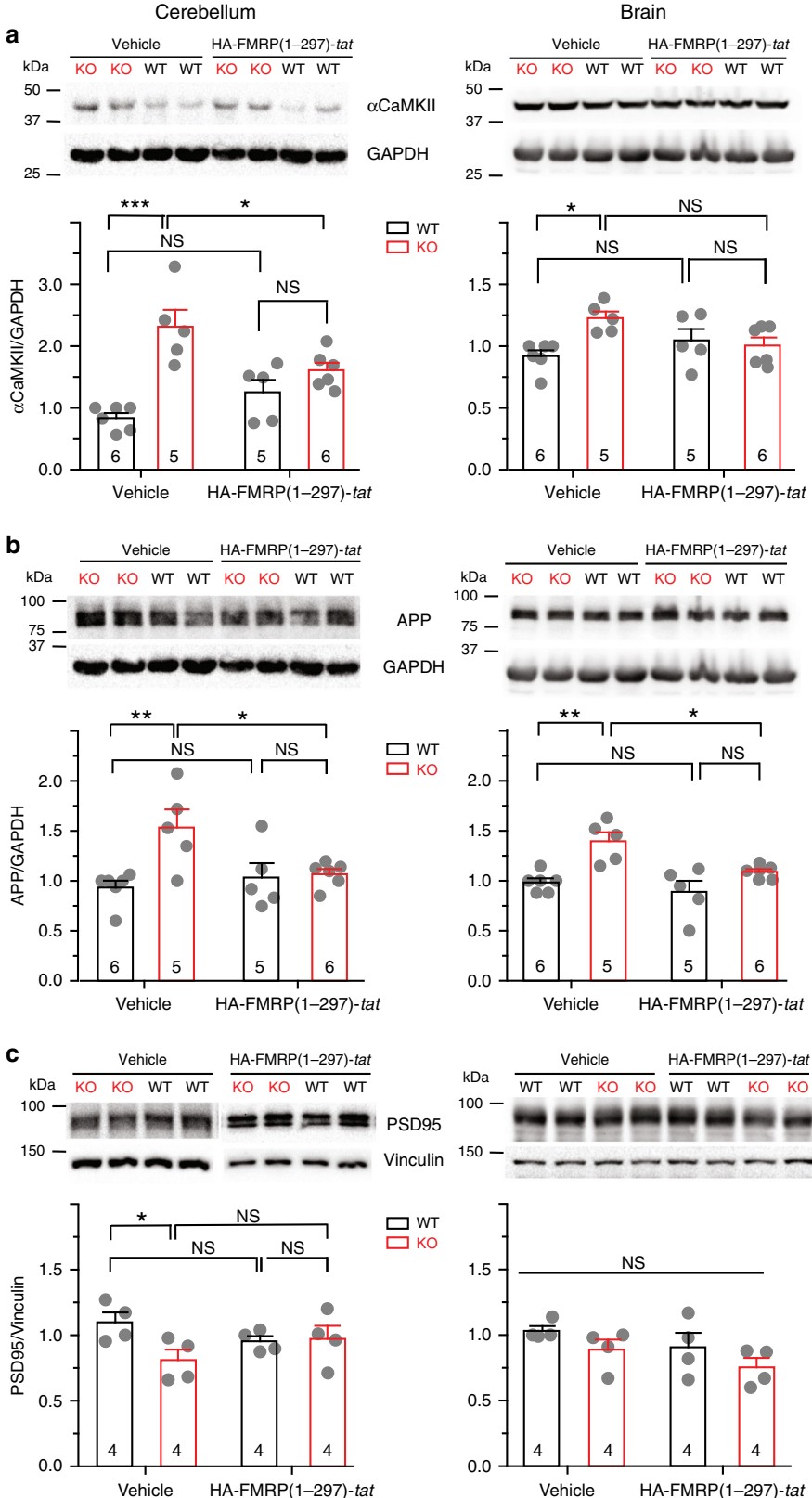

doublet discrimination were applied to screen for single neurons and assess the percentage of live vs dead cells in the population (Supplementary Fig. 5).

**Cerebellar granule cell electrophysiology**. Whole-cell patch recordings were obtained from mouse cerebellar granule cells of lobule 9 near the boundary between mossy fiber input and the granule cell body layer in sagittal tissue sections. Recordings were obtained using a Multiclamp 700B amplifier, Digidata 1440A and

pClamp 10.5 software and digitized at 40 kHz. Glass pipettes of 1.5 mm O.D. (A-M Systems) pulled using a P-95 puller (Sutter Instruments) had 4–8 MΩ resistance. For whole-cell voltage clamp recordings, series resistance was compensated to at least 70% and leak was subtracted offline in pClamp 10 software. For current clamp recordings cells with lower than −55 mV resting membrane potential were accepted, and less than 50 pA negative bias current applied to maintain a resting potential at ~−80 mV. Cells were allowed to equilibrate to the internal pipette solution for 3–5 min before recordings.

**Fig. 9 FMRP(1–297)-*tat* can restore levels of protein translation in *Fmr1* KO mice.** Shown are representative Western blots and mean bar-dot plots of the density of bands for the indicated proteins detected in cerebellum or brain samples from P21-30 WT or *Fmr1* KO mice 24 h following a single tail vein injection of either vehicle or 1.0 mg/kg HA-FMRP(1–297)-*tat*. Protein levels were normalized to GAPDH or vinculin. **a** *Fmr1* KO mice exhibit elevated levels of $\alpha$CaMKII that is significantly reduced 24 h following injection of HA-FMRP(1–297)-*tat* in samples from cerebellum but not in brain (two-way ANOVA analysis of the interaction between genotype and treatment: cerebellum, $F(1,18)$, interaction = 10.50, $p = 0.0045$, post hoc Tukey's test $p$(KO veh, WT veh) < 0.0001, $p$(KO veh, KO HA) = 0.0448; brain, $F(1,18)$, interaction = 7.161, $p = 0.0154$, post hoc Tukey's test $p$(KO veh, WT veh) = 0.0176, $p$(KO veh, KO HA) = 0.113). **b** Fmr1 KO mice exhibit elevated levels of APP that is significantly reduced by HA-FMRP(1–297)-*tat* injection in both cerebellum and brain (two-way ANOVA: cerebellum, $F(1,18)$, interaction = 6.176, $p = 0.0230$, post hoc Tukey's test $p$(KO veh, WT veh) = 0.0076, $p$(KO veh, KO HA) = 0.0438; brain, $F(1,18)$, interaction = 2.272, $p = 0.1491$, $F_{(1,18)}$, treatment = 8.204, $p = 0.0103$, $F_{(1,18),genotype} = 19.05$, $p = 0.0004$, post hoc Tukey's test $p_{(KO\ veh,WT\ veh)} = 0.0031$, $p_{(KO\ veh,KO\ HA)} = 0.0293$). **c** *Fmr1* KO mice exhibit reduced levels of PSD95 protein in cerebellum but not in brain (cerebellum, two groups of lysates ran on one blot, two-sample $t$ test, $t(6) = -2.636$. $p = 0.039$; brain, all four groups of lysates ran on one blot, two-way ANOVA analysis, $F_{(1,12),interaction} = 0.00417$, $p = 0.950$, $F_{(1,12),treatment} = 2.820$, $p = 0.119$, $F_{(1,12),genotype} = 3.630$, $p = 0.0810$, no significant difference among two-group comparisons). Samples after injection of HA-FMRP(1–297)-*tat* showed no significant difference between WT and *Fmr1* KO cerebellum (two-sample $t$ test, $t(6) = 0.174$. $p = 0.867$). Average values are mean ± s.e.m, and sample values identified at the base of bar plots represent the number of animals. *$p < 0.05$, ***$p < 0.001$. NS not significant. Source data are provided as a Source Data file.

Voltage-clamp recordings of Kv4 potassium current used an internal electrolyte of (mM): 140 KCl, 10 HEPES, 2.5 MgCl$_2$, 0.1 EGTA, pH 7.3 via KOH, with 5 di–tris-creatine phosphate, 2 Tris-ATP, and 0.5 Na-GTP added from fresh frozen stock each day. The external medium contained 30 μM CdCl$_2$ to block HVA calcium channels together with a 2 mM CsCl, 5 mM TEA, 1 μM TTX, glutamate receptor blockers DL-AP5 (25 μM) and DNQX (10 μM), and the inhibitory synaptic blockers picrotoxin (50 μM) and CGP55845 (1 μM). In recordings of Kv4 current before and after mossy fiber TBS the internal solution was composed of (mM) 140 KCl, 10 HEPES, 2.5 MgCl$_2$, 0.15 BAPTA, 0.05 CaCl$_2$, with 200 μM QX-314 internal to replace external TTX, and 5 di–tris-creatine phosphate, 2 Tris-ATP, and 0.5 Na-GTP added from fresh frozen stock each day, pH 7.3 via KOH. The external medium had no calcium channel blockers, but contained 2 mM CsCl, 5 mM TEA, and the inhibitory synaptic blockers 50 μM picrotoxin and 1 μM CGP55845. Kv4 or Cav3.1 activation and inactivation plots were calculated from currents evoked from a holding potential of −110 mV stepped in 10 mV (1000 ms) increments to 60 mV, followed by a return step to −30 mV (500 ms) as the test potential[8,9]. For tests on the involvement of protein translation on the shift in Kv4 Vh with LTP, 20 μM anisomycin was pre-applied to tissue slices for 2 h at RT and throughout recordings.

Current clamp recordings used an internal solution modified from Gall et al.[53]: 126 K-gluconate, 4 NaCl, 15 Glucose, 5 HEPES, 1 MgSO$_4$, 0.15 BAPTA, 0.05 CaCl$_2$, pH 7.3 via KOH, with 5 di–tris-creatine phosphate, 2 Tris-ATP, and 0.5 Na-GTP added from fresh frozen stock each day. The external medium contained 50 μM picrotoxin and 1 μM CGP55845. Mossy fibers were stimulated with a concentric bipolar electrode (Frederick Haer, CBCMX75 (JL2)) placed near the border of mossy fiber input and the granule cell layer using a Digitimer stimulus isolation unit (0.3 ms pulse). The stimulation protocol was the same as in Sola et al.[54], using a TBS pattern (8 bursts of 10 impulses at 100 Hz, 250 ms interburst interval) at a stimulus intensity that initially evoked a threshold EPSC or EPSP from a holding potential of −80 mV. For current-clamp recordings, TBS was delivered from a holding potential of ∼−65 mV. For voltage-clamp recordings of Kv4 current the TBS was paired with a postsynaptic voltage step from −70 to −40 mV for the duration of the TBS synaptic train (3 s total) as in D'Angelo et al.[55]. While TBS in current-clamp recordings used only synaptic stimulation. A 5 min period was used to establish baseline synaptic amplitudes before presenting TBS and EPSPs were recorded at 0.1 Hz to monitor LTP. EPSP amplitudes were calculated only for cases subthreshold to spike discharge and LTP was measured with respect to the mean value of all control records of baseline EPSP amplitude.

**FMRP and fusion of *tat* peptide to FMRP.** Recombinant FMRP(1–297) protein (H000023332-P01, Novus Biologicals, Oakville, ON) was added to the pipette before recordings or infused into the electrode during a recording (2PK+ Perfusion system; ALA Scientific, Farmingdale, NY). pTrc-HisA-*tat* vector was a modified pTrc-His A vector (Invitrogen,V360-20, Ottawa, ON) in which the Xpress epitope was replaced by an 11 aa *tat* sequence (YGRKKRRQRRR). The FMRP(1–297)-*tat* construct was made by subcloning FMRP(1–297) cDNA from *Fmr1* cDNA (Origene, RC222699) into the pTrc-HisA-*tat* vector with the following primers: FMR1-1F (5′-CTAGCTAGCATGGAGGAGCTGGTGGTGGAA-3′) and FMR1-297R (5′-CGGGATCCTATTACTTTGCCTACTAAGTT-3′). The construct was transformed and expressed in BL21 *Escherichia coli* bacteria (Invitrogen, C601003) grown in LB medium supplemented with 1 mg/ml Ampicillin (Sigma-Aldrich) overnight. The bacterial culture was further induced with 500 μM IPTG for 4 h for protein production. We purified FMRP(1–297)-*tat* peptide from the bacteria cultures by using the Ni-NTA Fast Start Kit (Qiagen, 30600, Toronto, ON). Aliquots of the peptide were stored at −80 °C in Elution Buffer of Ni-NTA Fast Start Kit (50 mM Na–phosphate, 300 mM NaCl and 250 mM imidazole at pH 8.0) and a proteinase inhibitor tablet was added to prevent proteolysis (one tablet in 10 ml solution) (04693124001, Roche). Protein integrity was tested by Coomassie Blue

Staining following sodium dodecyl sulfate polyacrylamide gel electrophoresis (SDS-PAGE) and quantified with the Bradford Protein Assay (Bio-Rad) following purification and before application.

A control *tat* fragment was prepared from the FMRP(1–297)-*tat* construct by excising the FMRP(1–297) region and replacing it with the restriction enzyme sites (ELEIC) that existed in the original pTrc-HisA vector. A *tat*-peptide was then purified from BL21 bacteria and stored in the same way as the FMRP(1–297)-*tat* peptides for control studies. An HA-FMRP(1–297)-*tat* construct that included an HA-tag sequence (YPYDVPDYA) on the N-terminus side of FMRP(1–297)-*tat* was prepared by using specific primers: HA-FMRP(1-297)-F-NheI, 5′-CTAGCTA GCTACCCATACGATGTTCCAGATTACG CTATGGAGGAGCTGGTGGTGG AA-3′ and HA-FMRP(1-297)-R-BamHI, 5′-CGGGATCCTATTACTTT TGCCTA CAAGTT-3′. The HA-tagged FMRP(1–297)-*tat* peptide was purified from BL21 bacteria and stored in the same manner.

FMRP(1–297)-*tat* proteins were delivered once by tail vein injection to animals under transient isoflurane anesthesia by inhalation (Midmark, Ohio) at 0.2–2.0 mg/kg (approximately 0.1–1 μM in blood upon injection) in 0.9% NaCl to a 200 μl volume, with an equivalent dilution of Elution Buffer (see above) to prepare vehicle for control groups. Injected animals were provided at least 1 h recovery time before any behavioral tests or 2 h before in vitro slice experiments were conducted.

**Heterologous expression in tsA-201 cells.** tsA-201 cells were maintained in DMEM supplemented with 10% heat inactivated fetal bovine serum and 1% pen–streptomycin at 37 °C under 5% CO$_2$. The calcium phosphate based method was used to transiently transfect cDNA[20]. Cells were washed with fresh medium 16–18 h after transfection and then transferred to 32 °C under 5% CO$_2$ for 1–2 days. Human Kv4.3 cDNA and human Cav3.1 cDNA was subcloned into the expression vector pCDNA3.1⁻ (Invitrogen). Human FMRP cDNA in pCMV6-Entry vector was obtained from OriGene and cDNA of both full length and the 1–297 fragment of FMRP were subcloned into mKate-PCDNA3.1⁻. In electrophysiology studies cells were transfected as indicated with cDNA of Cav3.1 or Kv4.3, eGFP cDNA was added (1 μg) to identify cells with successful transfection. FRET studies used 1 μg cDNA of GFP-Cav3.1, GFP-Kv4.3, and FMRP(1–297)-mKate or FMRP-mKate (full length). Transfected cells were incubated at 37 °C (5% CO$_2$) for 24 h and then transferred to a 32 °C incubator (5% CO$_2$) for 48–72 h prior to tests.

**tsA-201 cell electrophysiology.** Recordings of Kv4 current expressed in tsA-201 cells were carried out at 22 °C with an external solution comprised of (mM): 125 NaCl, 3.25 KCl, 1.5 CaCl$_2$, 1.5 MgCl$_2$, 10 HEPES, 5 TEA, 2 CsCl and 10 D-Glucose (pH adjusted to 7.3 with NaOH). Pipettes were filled with a solution comprised of (mM) 110 potassium gluconate, 30 KCl, 1 EGTA, 5 HEPES, and 0.5 MgCl$_2$, pH 7.3 via KOH, with 5 di–tris-creatine phosphate, 2 Tris-ATP, and 0.5 Na-GTP added from fresh frozen stock each day. Electrode solution could be infused with drugs using the 2PK+ System (ALA Scientific). To record Cav3.1 current pipettes were filled with a solution comprised of (mM) 100 CsCl, 10 EGTA, 10 HEPES, 2.5 MgCl$_2$, pH 7.3 with CsOH, with 5 di–tris-creatine phosphate, 2 Tris-ATP, and 0.5 Na-GTP added from fresh frozen stock each day. For Cav3.1 recordings the external medium contained (mM) 130 CsCl, 10 HEPES, 1 MgCl$_2$, 2 CaCl$_2$, pH 7.3 with CsOH.

**Fluorescence resonance energy transfer (FRET).** tsA-201 cells were plated onto poly-D-lysine coated 35 mm glass bottom culture dishes (World Precision Instruments, Sarasota, FL)[56]. Cells were transiently transfected with GFP-Cav3.1, GFP-Kv4.3, FMRP(1–297)-mKate or FMRP-mKate constructs to use as donor–acceptor fluorescent pairs[56]. On the experimental day DMEM was replaced with imaging medium comprised of (mM): 148 NaCl, 3 KCl, 10 HEPES, 3 CaCl$_2$, 10 D-Glucose, 1 MgCl$_2$ (pH 7.3 with NaOH) at 25 °C. Cells were examined on a Nikon Eclipse

C1Si spectral confocal laser-scanning microscope using a 40×/1.3NA oil immersion objective. Laser lines of 457 nm were used to excite GFP and 561 nm to excite mKate, with emission spectra recorded between 400 and 750 nm. Spectral images were linearly unmixed offline using ImageTrak software[56].

**Co-immunoprecipitation and Western Blotting**. For protein biochemical tests we define lysates prepared from whole brain as corresponding to all brain regions including cerebellum and brainstem, Brain to tissue that has cerebellum and hindbrain removed (Fig. 9), and cerebellum to isolated cerebellum. In no case was olfactory bulb tissue included. All coIP and blotting tests were conducted with the HA-FMRP(1–297)-*tat* construct for immunodetection. For coIP tests between FMRP and Cav3.1 and/or Kv4.3 and HA tag experiments, Whole brains (ranging 350–500 mg) were dissected out from P30 to P50 mice after isoflurane anesthesia and homogenized in a lysis buffer containing (mM): 150 NaCl, 50 Tris, 2.5 EGTA, 1% NP-40, pH 7.5, phosphatase inhibitor (P5726, Sigma-Aldrich) and proteinase inhibitor (04693124001, Roche). For protein level regulation tests (Fig. 9), brain and cerebellar lysates were prepared separately from WT or KO mice injected with FMRP(1–297)-*tat* or vehicle and homogenized in the same lysis buffer. The homogenates were then centrifuged at 13,000g for 10 min at 4 °C. Supernatants were collected and concentrations were measured using the Bradford Protein Assay (Bio-Rad) to provide 500 µg total protein for immunoprecipitation (IP) with rabbit anti-Cav3.1[56], rabbit anti-Kv4.3 (ab 65794, Abcam), or rabbit normal IgG (as a control) (ab 172730, Abcam) (antibodies in mixtures 40 µg/ml) and immuno-blotted with a mouse antibody against FMRP N-terminus (1:1000, ab230915, Abcam) (Fig. 2a). Whole brain lysates of *Fmr1* KO mice injected with HA-FMRP (1–297)-*tat* underwent IP with mouse anti-HA antibody (Abcam, ab130275) and mouse normal IgG (as control) (Abcam, ab 37355) (40 µg/ml) and were immuno-blotted with rabbit polyclonal anti-Cav3.1 and rabbit polyclonal anti-Kv4.3 antibodies (1:1000) (Fig. 5c). IP samples were incubated with antibodies for 2 h (4 °C) before the mixtures were incubated with 30 µl Protein G beads (Life Technologies) at 4 °C overnight. Beads were washed four times with lysis buffer by centrifugation and resuspension. The immuno-complexes were boiled at 95–100 °C for 5 min with 10 µl sample buffer diluted from 4× sample buffer (mM) 100 Tris, 100 2-mercaptoethanol, 4% SDS, 0.02% bromophenol blue, 20% glycerol, pH 6.8. IP samples in Fig. 2 (30 µl) and Fig. 5c (30 µl) and tissue homogenates (50 µg total protein) as input were loaded on 6–12% tris-glycine gel and resolved using SDS-PAGE. Samples were transferred to 0.2 µm PVDF membrane (Millipore) and probed with primary antibodies overnight at 4 °C, followed by goat anti-mouse (1:3000, 62–6520, Invitrogen) or donkey anti-rabbit (1:5000, NA-9340V, GE healthcare) HRP-conjugated secondary antibodies. Blot images were taken with a ChemiDoc imager and protein densities were analyzed with Image Lab (Bio-Rad). Other primary antibodies used in western blot include: rabbit polyclonal anti-APP (1:1000, A8717, Sigma-Aldrich), rabbit polyclonal anti-αCaMKII (1:3000, ab103840, Abcam), mouse monoclonal anti-PSD95 (1:1000, MABN68, Sigma-Aldrich), rabbit monoclonal anti-GAPDH (1:3000, 2118s, Cell Signaling Technology), and mouse monoclonal anti-vinculin (1:1000, SAB4200729, Sigma-Aldrich).

**Immunostaining**. Tissue for immunohistochemistry was obtained from P30 to P60 *Fmr1* KO or WT mice[8]. Animals were anesthetized by isoflurane inhalation until unresponsive to ear pinch and then perfused intracardially with 20 ml 0.1 M phosphate-buffer (PB, pH 7.4) followed by 20 ml 4% paraformaldehyde (PFA, pH 7.4) at RT. Brains were stored in 4% PFA at RT for 1 h and then overnight at 4 °C. Sagittal sections of 50 µm thickness were cut by vibratome (Leica VT1000 S, Germany) in PB. Tissue sections were blocked with a solution containing 10% normal goat serum, 0.2% DMSO and 0.1% TWEEN-20 and reacted overnight under 4 °C with the following primary antibodies diluted in working solution: rabbit monoclonal anti-FMRP C-terminal antibody (1:200, Cell Signaling Technology, 7104S), mouse monoclonal anti-HA (1:300, Abcam, ab130275), and chicken polyclonal anti-MAP2 (1:500, Abcam, ab92434) in dual labeling experiments. Primary antibodies were omitted in control sections for each experimental series. After washing in PB, sections were exposed for 1–2 h at RT to AlexaFluor 594-conjugated goat anti-mouse IgG (1:1000, Invitrogen, A11032) or AlexaFluor 488-conjugated goat anti-chicken IgG (1:1000, Invitrogen, A11039) and AlexaFluor 594-conjugated goat anti-rabbit IgG (1:1000, Invitrogen, A32740). Sections were washed 3 × 10 min in PB, mounted in Molecular Probes gold antifade medium and stored at −20 °C. Immunocytochemistry in granule cell culture was performed with the same process with minor modifications. Briefly, granule cell cultures were fixed in 4% PFA followed by 3 washes in PBS. Cells were permeabilized in PBS containing 0.2% Triton X-100 and blocked with 3% BSA in PBST buffer (0.1% Tween-20 in PBS). Imaging was conducted on a Zeiss Axioimager (Zen software) with Colibri LED illumination. Exposures were constrained to those used for control sections and post processing was restricted to equivalent adjustments of brightness/contrast for qualitative comparisons (Zen, Photoshop). Tiled montage images of cerebellar tissue sections were compiled from 80 to 90 images at 20× magnification (Zen).

**Open field test**. Different parameters of animal motion were quantified through an OFT using an overhead acA1300-60 hm camera (Basler, DE) and Noldus Ethovision XT 13 software (Leesburg VA). Male mice of P55-P100 age were placed in a square plexiglass chamber of 38 cm × 38 cm × 30 cm dimension and recorded during free movement for 30 min. A 4 × 4 grid pattern was applied to quantify different parameters during the OFT. HA-FMRP(1–297)-*tat* or vehicle were administered into animals as mentioned above. Analysis of total distance and duration of movement was conducted using Noldus Ethovision XT 13 and velocity was calculated by dividing total distance with duration for moving.

**Data analysis and statistical methods**. Inactivation curves were fitted according to the Boltzmann equation: $I/(I - I_{max}) = 1/(1 + \exp((V_h - V)/k))$, where $V_h$ is the half inactivation potential and $k$ is the slope factor. Activation curves were fit according to the Boltzmann equation: $G/(G - G_{max}) = 1/(1 + \exp((V_a - V)/k))$, where $G$ is calculated with equation $G = I/(V - V_{rev})$, $V_a$ is the half activation potential and $k$ is the slope factor. Inactivation and activation plots were constructed using Origin 8.0 (OriginLab, Northampton, MA). All figures were prepared and statistical analysis were performed using OriginPro 8 or GraphPad Prism 6, and Adobe Illustrator CC 2018 software.

Statistical significance was determined using a two-sample Student's *t* test for distinct samples, and a paired-sample Student's *t* tests for the same samples under different conditions. One-way ANOVA followed by post-hoc comparison (Tukey's multiple comparison) was used to analyze the statistical significance of more than two groups. In the OFT (Fig. 8) and protein translation test (Fig. 9), two-way ANOVA was applied to analyze the effect of genotype and treatment and their interactions, when necessary one-way ANOVA followed by post-hoc Tukey's comparison was applied to test the significance of different treatments in *Fmr1* KO mice. Normality of data was tested and there was no exclusion of outlier data points. Two-tailed analysis was chosen for all statistical tests.

**Reporting summary**. Further information on research design is available in the Nature Research Reporting Summary linked to this article.

## Data availability
All relevant data are available without restriction from the authors, with the original values for data underlying Figs. 1–3, 5–9, Supplementary Table 1, and Supplementary Figs. 1–4 provided as a Source Data file. Figures are also available to the public through a link on Figshare: 10.6084/m9.figshare.12132864. No accession codes are required.

## Code availability
ImageTrak software used for FRET imaging measurements is freely available at: http://www.ucalgary.ca/styslab/imagetrak. P.K. Stys, University of Calgary.

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

## Acknowledgements

We gratefully acknowledge Dr. Q. Long for advice and discussions on statistics, and J. Forden, M. Kruskic, R. Tobias, and Y.P. Liu for expert technical assistance. This work was supported by grants by a SFARI Explorer grant (R.W.T.), and grants from the Canadian Institutes for Health Research (R.W.T., J.M. Rho, and G.W.Z.) and Fragile X Canada/FRAXA (R.W.T.). Salary support was provided by the Alberta Children's Hospital Foundation (N.C.), and Postdoctoral Fellowships from the Hotchkiss Brain Institute, Cumming School of Medicine and Fragile X Canada/FRAXA (X.Z.), University of Calgary Eyes High (G.S.), and Alberta Innovates—Health Solutions (AI-HS) (G.S., F.-X.Z.). G.W.Z. holds a Canada Research Chair.

## Author contributions

X.Z., R.W.T., H.A., and N.C. designed the experiments, X.Z. and G.S. conducted electrophysiology experiments, X.Z. and H.A. protein biochemical tests and FRET imaging, N.C., E.S., and X.Z. behavioral tests and analysis, H.A. and F.-X.Z. prepared fluorescent tagged and *tat* constructs, R.W.T., X.Z., N.C., and H.A. wrote the paper, R.W.T. and N.C. supervised the study. G.W.Z. and J.R. provided necessary reagents and supplies.

## Competing interests

The following patent application exists: Applicant: UTI Limited Partnership. Inventors: Turner, Raymond W Application Number: PCT/CA2019/000001 Status: Application Specific aspect of the paper covered by the patent application: Use of tat-FMRP peptides to treat Fragile X Syndrome.
