## [Peer Review File · Nature Communications]

Reviewers' Comments:

Reviewer #1:

Remarks to the Author:

This manuscript describes the loss of long-term potentiation (LTP) at the mossy fiber-granule cell synapse in a mouse model of Fragile X syndrome. This loss of LTP is restored by the infusion of a peptide corresponding to the first 297 amino-acids of FMRP, in the patch pipette. In the cerebellum, LTP at the mossy fiber-granule cell synapse depends on the modulation of the Cav3-Kv4 channel complex and the authors link the lack of LTP in FMRP KO granule cells to a loss of modulation of Kv4 channels. The modulation of Kv4 channels is also restored by the infusion of the N-terminal fragment of FMRP. The authors then showed a novel interaction between FMRP, Cav3 and Kv4 channels. The authors also show that a FMRP(1-297)-tat peptide, injected in vivo, can be expressed in cerebellar granule cells and can restore LTP at the mossy fiber-granule cell synapse and can reduce hyperactivity in adult FMRP KO mice.

The results presented here demonstrate another critical function for FMRP and a very exciting therapeutic avenue to treat fragile X syndrome. The experiments presented here are convincing. However, the impact of the work could be increased as described in the specific comments below.

LTP at the mossy fiber-granule cell synapse lost in FMRP KO mice:

Data presented in Fig1a: the average value (+ SEM) for the initial increase in EPSP amplitude 5 min after TBS in WT should be stated rather than the max value ("up to 250%"). A numerical value for the spike occurrences (firing probabilities) have to be stated even if the difference looks obvious between before and after TBS in wt; these values are required to compare wt condition vs Fmr1-/-y and Fmr1-/-y + FMRP(1-297).

Fig.1 title: "Mossy fiber LTP and modulation of the Cav3-Kv4 complex depends on FMRP", the title is misleading since there is no evidence of the involvement of Cav3 channels/or the impact on Cav3 channels in Fmr1 KO mice. Would the use of a T-type blocker give some indication on whether Cav3 channel activity is modified in the FMRP KO background?

Later in the manuscript (figure 3f), the authors show no effect of TTA-P2 on Kv4 current recorded from FMRP KO granule cells treated with FMRP(1-297). How do they reconcile these data?

FMRP restores mossy fiber LTP in FMRP KO mice:

Full length FMRP is about 632 aa (depending on splicing) so FMRP(1-297) fragment is not that "short"!

Back to spike occurrences (firing probabilities): the spike occurrence before TBS in Fig1c (Fmr1-/-y + FMRP(1-297)) looks much higher than in Fig.1a (wt)! After TBS, even though it is clear that there is an increase in firing probability (looks even higher than wt!), it is important to have a numerical value to be able to compare it with the wt. For the EPSP amplitude after TBS, the "increase" (19%, n= 6, P=0.78) does not look very convincing.

Fig.1h,i: FMRP is a RNA binding protein that modulates protein translation (Darnell et al. 2011 Cell). FMRP(1-297) still contains a KH1 domain, is the effect observed on Kv4 depends on protein translation? The use of a protein synthesis inhibitor could be informative on the mechanism of action of FMRP (see Fig 1 in Deng et al. 2013 Neuron).

FMRP associates with Cav3.1 and Kv4.3 channels.

Fig. 2a: it is good practice to run input and co-IP products on the same gel to show that the identified bands migrate at the same size. As a negative control, an "irrelevant" primary antibody (or just rabbit IgG) should be used instead of beads alone. Information about the amount of tissue used to prepare the homogenates, the amount of total protein used for the IP, the amount of protein from the lysate loaded on the gel. These details would give the reader some quantitative indication on level of expression of the proteins in the lysate and compare it to the amount that is pulled-down and give information on the quality of the interaction between the binding partners. Cav3 and Kv4.3 exists as a complex, would FMRP be able to pulldown both channels at the same time?

Fig.2.b,d: Whereas GFP-Cav3.1 and GFP-Kv4.3 are expressed throughout the cells, FMRP(1-297)-mKate seems to be expressed in aggregates. On the FRET images (excitation at 457nm) it is

difficult to distinguish GFP signal from FRET signal, it would help to have GFP-tagged channels with GFP filter on their own as it was shown in Asmara et al. 2017 Mol Brain (Fig.2). Supplemental Figure 1 with full length FMRP looks much better. Maybe this figure should be move to the main text.

Where is the interaction occurring, plasma membrane and/or inside the cell? The specificity of the FRET signal between channels and FMRP could be challenged using untagged FMRP(1-297). "data not shown" corresponding to negative control should be included at least as supplemental data.

Direct interaction between FMRP and Slack potassium channel and Cav2 channels have been previously described and should be further discussed in terms of interaction domains and modulation (biophysical properties vs cell surface expression).

FMRP modulates Cav3.1 and Kv4.3 channels

The data presented in the top part of the table 1 are already described in Fig 3a to 3d, and the Kv4 Va values for Fmr1-/- (with or without FMRP1-297) could easily be included in the legend of the Figure 3.

For Cav3.1 currents, it is not clear whether the reduction of current amplitude is related to a reduction of current density.

For Kv4.3, it is also a bit confusing because the authors describe a reduction in Kv4.3 current amplitude in tsA cells with FMRP(1-297) infusion compare with control (this reduction being likely due to a leftward shift of the Vh) so unless the cell capacitance is decreasing during the recording the currenty density (pA/pF) should be reduced! And it is a bit surprising that the authors are using data obtained from granule cells (supplemental Fig.2) to conclude that the reduction in current amplitude recorded in tsA cells expressing Kv4.3 channels is not due to a reduction in current density. Is there a bit of confusion between current density and channel density (number of functional protein) at the plasma membrane?

A FMRP(1-297)-tat peptide rapidly crosses the blood brain barrier to access central neurons.

Considering the fact that the FMRP(1-297)-tat peptide used at 500nM is no longer effective to reduce hyperactivity after 24h (figure 7), the authors should check that the tat peptide is still detectable in the cerebellum.

FMRP(1-297)-tat restores mossy fiber-granule cell LTP.

The authors tested the effect of 100 nM and 500 nM tat peptide injection on LTP. Only 500nM had an effect. However, there is no evidence that using a 100 nM tat peptide injection, HA is detectable in the cerebellum.

FMRP(1-297)-tat reduces hyperactivity in Fmr1-/- mice.

As the effect of the FMRP tat peptide appears to be transitory, it would be interesting to know if repeated injection could maintain the effect of FMRP-tat (500 nM).

Minor comments:

In the discussion:

Rotarod (instead of rotorod)

"..., while 1 microM injection was less effective." This sentence should be re-phrased as the results obtained using 1 microM were not significantly different from the control.

"Yet injecting 500 nm..." should read "nM"

Reviewer #2:

Remarks to the Author:

This manuscript describes a novel approach that the authors indicate could be used to treat FXS. The results also add new findings to the body of literature on the effects of FMRP on voltage-gated channels, particularly potassium channels. Most importantly, the authors have studied a novel approach to FMRP replacement in the CNS by examining the effects of FMRP1-297-tat in vitro and in vivo.

Critique

1. Overall, the paper is well-written. However, the first 2 sections of the results section and the

data in Fig. 1 are a bit difficult to follow because of the way the data in Fig. are divided in the text. Also, the panels in the figure appear to be jumbled. Combining the results shown in Fig. 1 into a single test subsection would improve in the flow. More importantly, the effect of FMRP1-297 on LTP amplitude shown in Fig. 1C appears minimal. $p = 0.78$. It would be useful to make an explicit statement as to the interpretation of the effect shown in 1c.

2. The set-up in Fig. 2a is also a bit confusing because of the vertical arrangement of the WB and the use of the word "beads"; this should be changed. Also, why are the IP target proteins not shown in the figure?

3. Some of the data and text describing Fig. 3 are also hard to follow and some of the panels appear out of order or not described in a logical manner in the results text (again, a flow issue in the text and/or figure). What was infused in the control experiments?

4. For the results text in Fig. 4 what is the meaning of "...within the range of high efficiency transport" and what does it refer to? Are the data in table 1 from cerebellar slice recordings? If so what is the age of the mice? How long was the FMRP 1-297 incubated with the slice? This info should be added to the table legend.

5. It is stated "We tail-vein injected 500 nM FMRP1-297-tat...". What was the actual amount injected (also volume and length of infusion time)? Most of the results are from 30 min after the injection. How does the level of expression compare at the 12 hour time measured, and at later times (e.g. after 3 days and 1-2 weeks)?

In other words, what is the time frame of FMRP1-297-tat expression as determined by IF and WB?

6. How old and what sex were the animals that were injected via the tail vein?

7. The motor activity data in Fig. 7 are quite interesting and the analysis is comprehensive. It is commendable that the authors also studied effects in wt mice and tested different doses. The results show a corrective effect on motor hyperactivity and velocity, but only at a single dose and only immediately after tail vein infusion of FMRP1-297-tat. As noted above, the manuscript would be improved by quantitating FMRP1-297-tat in the CNS at appropriate time points after infusion.

8. In the discussion it is stated that:

"Another strategy is to use a tat-conjugate peptide, which we introduced by intravenous administration. An advantage to this is the ability to define the initial concentrations applied, avoiding the negative effects of overexpressing FMRP." This statement is somewhat misleading in that simply knowing how much is injected into the tail does not in itself avoid negative effects of over-expressing the protein. In fact, very little data were shown that addressed this point in terms of pan-CNS expression after peripheral administration.

9. Overall, an important observation in this study is that rapid reversal of synaptic plasticity and other abnormal parameters in *fmr1* ^{-/-} mice after infusion of FMRP1-297-tat. This is an important confirmation that the phenotype of FXS can be reversed during or after the developmental stage. The possible short term nature of the effects might limit the development of this approach in terms of a therapeutic. Are there any work-arounds that address this key issue that could be tried in the future and mentioned in the Discussion?

Minor comments

FMRP1-297 is described as a "short fragment of FMRP" but cannot be described as such because it is about one half of the protein.

What is the "full-length FMRP-mKate construct"?

Reviewer #3:

Remarks to the Author:

The authors present an interesting approach to restoring functions affected in Fragile X syndrome. Injection of a blood-brain-barrier crossing peptide containing a short N-terminal fragment of FMRP restores specific neural circuit functions in a mouse model for Fragile X (*Fmr1*-/*y* mice), which rest on the complex forming capacity of this fragment with Cav3 and Kv4. The authors argue that this demonstration may provide the basis for future attempts to reduce FXS symptoms using peptide fragment injections. A link to behavioral output is also provided: peptide delivery reduces hyperactivity, which is seen in these mice.

This paper is of interest, but of very limited scope. The question does arise to what degree this is a useful proof of principle demonstration. The interaction of FMRP with the Cav3-Kv4 complex is certainly important, but there is currently no evidence that interruption of this interaction contributes to any symptom in FXS (never mind that there is not even an established correlation between granule cell excitability and hyperactivity). Thus, the rescue effect, if limited to this N-terminal fragment and its functional roles, might be marginal.

There are two possible routes the authors could go to make this study more impactful. First, to study effects on sensory processing in mice that result from FMRP deletion from cerebellar granule cells. This approach would involve the generation of granule cell specific FMRP deletion, and demonstration of sensory rescue in behaving mice.

Second, and alternatively, the authors might want to show that a similar effect can be observed on protein translation with a focus on peptides mimicking protein fragments involved in CYFIP1 and eIF4E interactions through which FMRP controls translation. In the context of this paper –that makes the point of the general usefulness of peptide injection strategies – it would be sufficient to test expression levels of specific marker proteins, without further behavioral testing.

It is true that both suggestions go a bit beyond the scope of the manuscript as submitted. That is, because as it is, the study is appealing from an experimental perspective, but will be of little consequence. The relation of FMRP function in excitability control toward FXS phenotypes is too speculative, there is not a single behavioral phenotype that has been linked to it. This also holds true for the observed hyperactivity as there is no evidence that the injected peptide acts through its effects on the Cav3-Kv4 complex in granule cells. It also remains possible that the N-terminal fragment interacts with other, unrelated protein complexes.

We thank the reviewers for their constructive comments and suggestions on how to improve the manuscript. We believe we have been able to answer virtually all requests, and completed new experiments tracking the distribution and presence of FMRP-tat in the brain over time, and testing its effects on protein translation. Both tests prove to add considerable insight into the potential for this compound to represent a potential therapeutic avenue to alleviate the symptoms of Fragile X Syndrome. We hope the reviewers will be satisfied with our revisions, and provide guidance to all changes in the manuscript using Blue text.

Ray W Turner
Cumming School of Medicine
University of Calgary

Reviewers' comments:

Reviewer #1 (Remarks to the Author):

This manuscript describes the loss of long-term potentiation (LTP) at the mossy fiber-granule cell synapse in a mouse model of Fragile X syndrome. This loss of LTP is restored by the infusion of a peptide corresponding to the first 297 amino-acids of FMRP, in the patch pipette. In the cerebellum, LTP at the mossy fiber-granule cell synapse depends on the modulation of the Cav3-Kv4 channel complex and the authors link the lack of LTP in FMRP KO granule cells to a loss of modulation of Kv4 channels. The modulation of Kv4 channels is also restored by the infusion of the N-terminal fragment of FMRP. The authors then showed a novel interaction between FMRP, Cav3 and Kv4 channels. The authors also show that a FMRP(1-297)-tat peptide, injected in vivo, can be expressed in cerebellar granule cells and can restore LTP at the mossy fiber-granule cell synapse and can reduce hyperactivity in adult FMRP KO mice.

The results presented here demonstrate another critical function for FMRP and a very exciting therapeutic avenue to treat fragile X syndrome. The experiments presented here are convincing. However, the impact of the work could be increased as described in the specific comments below.

LTP at the mossy fiber-granule cell synapse lost in FMRP KO mice:

Data presented in Fig 1a: the average value (+ SEM) for the initial increase in EPSP amplitude 5 min after TBS in WT should be stated rather than the max value ("up to 250%"). A numerical value for the spike occurrences (firing probabilities) have to be stated even if the difference looks obvious between before and after TBS in wt; these values are required to compare wt condition vs Fmr1 KO and Fmr1 KO + FMRP(1-297).

We have removed the max value of EPSP amplitude indicated above in favor of staying consistent with our other measures of providing the mean and SEM values 10-15 min after TBS was delivered. This has now been entered into the Results section.

The probability of spike firing was calculated and added to Figure 1a-c and described in the Results for the mean values associated with baseline and post TBS recordings for the conditions of WT, Fmr1 KO, and Fmr1 KO + FMRP(1-297).

Fig.1 title: "Mossy fiber LTP and modulation of the Cav3-Kv4 complex depends on FMRP", the title is misleading since there is no evidence of the involvement of Cav3 channels/or the impact on Cav3 channels in Fmr1 KO mice. Would the use of a T-type blocker give some indication on whether Cav3 channel activity is modified in the FMRP KO background?

The reason we used the phrase Cav3-Kv4 complex is that our previous work has shown that these channels form a tight complex to enable calcium-dependent regulation of Kv4 Vh and availability (Anderson et al. 2010; Heath et al. 2014; Rizwan et al. 2016). The Kv4 current we recorded under our conditions was thus expected to include Cav3-dependent modulation. However, it is true that until we specifically tested Cav3 and Kv4 separately

for colP and FMRP-dependent regulation in Figs 2 & 3 the title could be misleading. We thus changed the title of Figure legend 1.

Later in the manuscript (figure 3f), the authors show no effect of TTA-P2 on Kv4 current recorded from FMRP KO granule cells treated with FMRP(1-297). How do they reconcile these data?

To clarify, we found that infusion of FMRP(1-297) into *Fmr1* KO granule cells produced a significant leftward shift in Kv4 Vh in the absence (Fig. 3c) or presence of TTA-P2 to block Cav3 calcium channels (Fig. 3d). We interpret this to indicate that the actions of FMRP(1-297) on Kv4 current can be produced independent of actual calcium flux across the plasma membrane (i.e. FMRP can bypass the established ability for a decrease in membrane calcium conductance to left-shift Kv4 Vh) (Anderson et al., 2010, 2013; Heath et al., 2014; Rizwan et al., 2016). While we do not dismiss the potential role of Cav3 calcium conductance and actions on the Cav3-Kv4 complex (now under further study in a separate project), this finding illustrates that the effects of FMRP(1-297) on Kv4 channels or other members of the complex may be sufficient to account for many of the results obtained upon infusion of this N-terminal segment of FMRP.

FMRP restores mossy fiber LTP in FMRP KO mice:

Full length FMRP is about 632 aa (depending on splicing) so FMRP(1-297) fragment is not that "short"!

We removed the descriptor for FMRP(1-297) as being "short" from the text.

Back to spike occurrences (firing probabilities): the spike occurrence before TBS in Fig1c (*Fmr1* KO + FMRP(1-297)) looks much higher than in Fig.1a (wt)! After TBS, even though it is clear that there is an increase in firing probability (looks even higher than wt!), it is important to have a numerical value to be able to compare it with the wt.

We feel that several factors contribute to these apparent differences. First, we can not assume that a change in granule cell excitability in the *Fmr1* KO animals only reflects a change in Kv4 properties. As mentioned later in the text, the differences we find between LTP induction for direct postsynaptic infusion of FMRP(1-297) compared to whole animal injection of FMRP-tat suggest that additional presynaptic factors must contribute to changes in excitability upon loss of FMRP (cf **Fig. 1c** and **Fig. 6**). Second, In the LTP experiments of Fig 1 we chose to include 3 nM FMRP(1-297) in our electrode. While we did not detect markedly different changes in baseline excitability we can not guarantee that this is the optimal concentration, or one that matches the concentrations achieved following tail vein injections. It is thus possible that granule cells of *Fmr1* KO mice exhibit a slightly higher level of excitability in response to synaptic input in the presence of internal FMRP(1-297). To help clarify these issues we have plotted the mean value of spike probability (binned at 1 min) for each condition in **Figs. 1 & 6**. This would support the reviewers' observation of a slightly higher level of firing probability in response to synaptic input in FMRP(1-297) infused into *Fmr1* KO mouse granule cells compared to WT. However, the ability for mossy fiber TBS to invoke a longer term additional increase in firing probability in the presence of FMRP(1-297) was confirmed for postsynaptic infusion (**Fig. 1c**) as well as extracellular FMRP(1-297)-tat application (**Fig. 6**).

For the EPSP amplitude after TBS, the "increase" (19%, n= 6, P=0.78) does not look very convincing.

We agree, and meant to indicate that when we apply FMRP(1-297) directly in the electrode the primary influence of TBS is on the probability of spike firing, but not EPSP amplitude. This has now been emphasized in Fig legend 1, and in the text with the mean value 10-15 min post TBS provided. We also changed the subtitle of this section in the Results to emphasize the selective effect of FMRP(1-297) infusion on spike firing.

Fig.1h,i: FMRP is a RNA binding protein that modulates protein translation (Darnell et al. 2011 Cell). FMRP(1-297) still contains a KH1 domain, is the effect observed on Kv4 depends on protein translation? The use of a protein synthesis inhibitor could be informative on the mechanism of action of FMRP (see Fig 1 in Deng et al. 2013 Neuron).

This is an interesting suggestion that we have now tested and added as **Supplemental Figure 1**. Here we found that applying 20 μ M anisomycin for 2 hrs before and during recordings had no significant effect on the TBS-induced

shift in Kv4 Vh in WT granule cells or cells in *Fmr1* KO animals when 3 nM FMRP(1-297) was included in the electrode. We thus conclude that the effects of FMRP(1-297) on Kv4 Vh as a mediator of postsynaptic elements of LTP do not reflect an induced change in protein translation, and state this in the Results and Discussion. Additional tests on the ability for tat-FMRP to restore altered protein levels is presented in **Figure 9**.

FMRP associates with Cav3.1 and Kv4.3 channels.

Fig. 2a: it is good practice to run input and co-IP products on the same gel to show that the identified bands migrate at the same size. As a negative control, an "irrelevant" primary antibody (or just rabbit IgG) should be used instead of beads alone. Information about the amount of tissue used to prepare the homogenates, the amount of total protein and antibodies used for the IP, the amount of protein from the lysate loaded on the gel. These details would give the reader some quantitative indication on level of expression of the proteins in the lysate and compare it to the amount that is pulled-down and give information on the quality of the interaction between the binding partners.

The additional lane for a negative control (IgG) has been added to these blots (**Figs. 2a, 5**). IP samples in Fig. 2 (30 μ l) and Fig.5 (10 μ l) and tissue homogenates (50 μ g total protein) as input were loaded on 6 -12% Tris-glycine gel and resolved using SDS-PAGE. These details on the protein loading have been provided in Methods section.

Cav3 and Kv4.3 exists as a complex, would FMRP be able to pulldown both channels at the same time?

To test this we attempted to reverse the coIP in **Fig. 2a** but the available N-terminal FMRP antibodies did not function well for immunoprecipitation (IP). Instead we used IP of the HA tag on *Fmr1* KO mice tail vein injected with HA-FMRP(1-297)-*tat* and detected a coIP with both Cav3.1 and Kv4.3 (**Fig. 5d**). Given the amount of protein available for these blots, detecting the coIP between HA-FMRP and Cav3.1 or Kv4.3 required a higher dose of tail vein injection of HA-FMRP(1-297)-*tat* (26 mg/kg) and IP for HA to enrich the sample. We note that this does not entirely establish whether there is a strong bond between FMRP and both Cav3.1 and Kv4.3, but the tests are valuable in validating an association of the injected HA-FMRP(1-297)-*tat* with the channels of interest to synaptic plasticity in the cerebellum.

Fig.2.b,d: Whereas GFP-Cav3.1 and GFP-Kv4.3 are expressed throughout the cells, FMRP(1-297)-mKate seems to be expressed in aggregates. On the FRET images (excitation at 457nm) it is difficult to distinguish GFP signal from FRET signal, it would help to have GFP-tagged channels with GFP filter on their own as it was shown in Asmara et al. 2017 Mol Brain (Fig.2). Supplemental Figure 1 with full length FMRP looks much better. Maybe this figure should be move to the main text.

As suggested we exchanged the full length FMRP vs FMRP(1-297) FRET tests between the main text and **Supplemental Fig. 2**. This also maintains a focus on the full length FMRP molecule that is tested by coIP from brain homogenates in Figure 2a. We also place more emphasis on the emission measurements in FRET since visualization of fluorescence is helpful but not reliable. If clumping does occur it did not alter the control emissions for FMRP(1-297)-mKate tests shown in a new **Supplemental Fig. 2**.

The specificity of the FRET signal between channels and FMRP could be challenged using untagged FMRP(1-297). "data not shown" corresponding to negative control should be included at least as supplemental data.

We have now provided a full **Supplemental Fig 2** testing any potential non-specific interactions between the tagged proteins and fluorophores.

Where is the interaction occurring, plasma membrane and/or inside the cell? Direct interaction between FMRP and Slack potassium channel and Cav2 channels have been previously described and **should be further discussed** in terms of interaction domains and modulation (biophysical properties vs cell surface expression).

We recognize that other studies have traced the molecular basis of interactions between FMRP and specific channels which is now stated in the Discussion. We do not yet have a specific answer as to where/how FMRP

interacts with the proteins potentially involved here beyond that it is restricted to the N-terminal component of FMRP. We realize this is important, and are proceeding to identify the molecular interactions between FMRP and each of Cav3, Kv4.2/4.3, KChIP3, and DPP subunits that all contribute to activity of this complex in a second study. We can say that given the speed at which FMRP(1-297) changed the biophysical properties of Cav3 or Kv4.3 channels in tsA-201 or granule cells, and the results with coIP and FRET that FMRP can be closely associated with the Cav3-Kv4 complex, suggesting biophysical effects are mediated at the membrane level. In support of this we state in the Results that FMRP(1-297)-tat did not alter the current density of Kv4 A-type current (pA/pF) (when tested from -110 mV), ruling out a change in channel surface expression (presented in **Supplemental Fig. 4**). We did not attempt to measure the density for Cav3 channels given an amplitude of <10 pA in WT mice. Our most recent tests with anisomycin now also show a lack of involvement of protein translation for FMRP(1-297) in restoring LTP (**Supplemental Fig. 1**). In the Results we now state that even with these associations we can not conclude direct interactions between FMRP and these channel subunits without further work given the potential action of any of the accessory proteins on channel function. In the Discussion we present how our findings stand in relation to previous work that have narrowed down the interaction sites.

FMRP modulates Cav3.1 and Kv4.3 channels

The data presented in the top part of the table 1 are already described in Fig 3a to 3d, and the Kv4 Va values for Fmr1 KO (with or without FMRP1-297) could easily be included in the legend of the Figure 3.

We removed Table 1 entirely and moved relevant values to the Results or Fig. Legend 3.

Current Density measurements

- For Cav3.1 currents, it is not clear whether the reduction of current amplitude is related to a reduction of current density.
- For Kv4.3, it is also a bit confusing because the authors describe a reduction in Kv4.3 current amplitude in tsA cells with FMRP(1-297) infusion compare with control (this reduction being likely due to a leftward shift of the Vh) so unless the cell capacitance is decreasing during the recording the current density (pA/pF) should be reduced!
- And it is a bit surprising that the authors are using data obtained from granule cells (supplemental Fig.2) to conclude that the reduction in current amplitude recorded in tsA cells expressing Kv4.3 channels is not due to a reduction in current density. Is there a bit of confusion between current density and channel density (number of functional protein) at the plasma membrane?

We have clarified in **Figure Legend 3** that when we measure a decrease in Cav3.1 or Kv4 current amplitude it is specifically from a holding potential of -70 mV (near resting potential) and with a test pulse to -30 mV (the peak voltage for Cav3 current) (Anderson et al., 2010; Heath et al., 2014; Rizwan et al., 2016) This is to assess the effects of experimental manipulations on Cav3 or Kv4 Vh over the membrane potential range relevant to a cell and spike output. Since there is a voltage shift in Kv4 Vh in granule cells over a range of potentials from ~-100 mV to -50 mV it is not possible to use a pA/pF measurement to assess potential changes in channel density. Instead, we assess any change in channel density at the membrane by stepping from -110 mV to +60 mV to activate the maximum current available. As shown in **Supplemental Fig. 4** these tests did not return evidence for a change in Kv4 channel density with either direct infusion of FMRP(1-297) during recordings or following tail vein injection of FMRP(1-297)-tat. While similar tests were attempted with Cav3 current, it was too small (<10 pA) to reliably measure.

A FMRP(1-297)-tat peptide rapidly crosses the blood brain barrier to access central neurons.

Considering the fact that the FMRP(1-297)-tat peptide used at 500nM is no longer effective to reduce hyperactivity after 24h (figure 7), the authors should check that the tat peptide is still detectable in the cerebellum.

FMRP(1-297)-tat restores mossy fiber-granule cell LTP.

The authors tested the effect of 100 nM and 500 nM tat peptide injection on LTP. Only 500nM had an effect. However, there is no evidence that using a 100 nM tat peptide injection, HA is detectable in the cerebellum.

To be more clear and to stay with convention we have changed all concentration references for animal injections to the values for mg/kg injected FMRP-tat compound. The intended initial plasma concentration of 500 nM thus equates to 1.0 mg/kg. We have found that the 1 mg/kg concentration for tail vein injection that is

effective at reducing hyperactivity (Fig. 8) and restores LTP (Fig. 7) and protein translation (Fig. 9) is below the limits of detection by standard Western blots from cerebellar or brain lysates, despite it being detectable by immunocytochemistry in tissue sections (Fig. 5a, b). This is supported by tests in cultured granule cells where Western blots could detect FMRP(-1-297)-tat directly applied to these cells, but only for concentrations ≥ 10 nM (not shown). Nevertheless, we have now been able to detect tail vein injected HA-FMRP(1-297)-tat in brain lysate by immunoprecipitating for HA and immunoblotting with an N terminal FMRP antibody (Fig. 5c). We can also detect FMRP immunolabel in tissue sections after tail vein injection of FMRP(1-297)-tat at 100 nM (Supplemental Fig. 5). We further added new tests to compare immunolabel intensity and quantify band density on colPs over time, revealing the presence, if not steady uptake, of FMRP-tat in brain and cerebellar lysate up to 24 hr time that dissipates by 48 hr (Fig. 5a,b).

FMRP(1-297)-tat reduces hyperactivity in Fmr1 KO mice.

As the effect of the FMRP tat peptide appears to be transitory, it would be interesting to know if repeated injection could maintain the effect of FMRP-tat (500 nM).

Yes, we agree. These tests are included as part of a Commercialization grant now in preparation as this will take a large breeding colony to support the required assessment of animals >P60.

Minor comments:

In the discussion:

Rotarod (instead of rotorod)

“..., while 1 microM injection was less effective.” This sentence should be re-phrased as the results obtained using 1 microM were not significantly different from the control.

“Yet injecting 500 nm...” should read “nM”

Done, and all values replaced with a mg/kg equivalent value.

Reviewer #2 (Remarks to the Author):

This manuscript describes a novel approach that the authors indicate could be used to treat FXS. The results also add new findings to the body of literature on the effects of FMRP on voltage-gated channels, particularly potassium channels. Most importantly, the authors have studied a novel approach to FMRP replacement in the CNS by examining the effects of FMRP1-297-tat in vitro and in vivo.

Critique

1. Overall, the paper is well-written. However, the first 2 sections of the results section and the data in Fig. 1 are a bit difficult to follow because of the way the data in Fig. are divided in the text. Also, the panels in the figure appear to be jumbled. Combining the results shown in Fig. 1 into a single test subsection would improve in the flow. More importantly, the effect of FMRP1-297 on LTP amplitude shown in Fig. 1c appears minimal. $p = 0.78$. It would be useful to make an explicit statement as to the interpretation of the effect shown in 1c.

- Panel configuration - we have attempted to clarify the text around Figure 1 by grouping the combined measures of V_h and V_a for the three animal groups (WT, Fmr1 KO and Fmr1 KO + FMRP) into a d, e, f configuration.
- Effects of FMRP(1-297) on EPSP amplitude in Fig 1c - as noted above, we have clarified the text to state there was no significant effect on EPSP amplitude for a postsynaptic infusion of FMRP(1-297) (as compared to FMRP-tat applied by tail vein injection - see Fig 6). We have also added spike probability plots that will help clarify the effects of postsynaptic FMRP(1-297) on cell excitability in current clamp mode.

2. The set-up in Fig. 2a is also a bit confusing because of the vertical arrangement of the WB and the use of the

word “beads”; this should be changed. Also, why are the IP target proteins not shown in the figure?

As described above for Reviewer 1, these experiments were repeated and reorganized in their presentation

3. Some of the data and text describing Fig. 3 are also hard to follow and some of the panels appear out of order or not described in a logical manner in the results text (again, a flow issue in the text and/or figure). What was infused in the control experiments?

- Panel configuration - We reduced the number of figure frames to group tests on Vh/Va together to clarify the presentation.
- Vehicle - The solution used for Vehicle has been clarified in the Methods section, and the injection of injecting just the tat epitope in vehicle is identified in Fig. 7e,f and 8a, d

4. For the results text in Fig. 4 what is the meaning of “...within the range of high efficiency transport” and what does it refer to? Are the data in table 1 from cerebellar slice recordings? If so what is the age of the mice? How long was the FMRP 1-297 incubated with the slice? This info should be added to the table legend.

- The phrase regarding “range of high efficiency transport” refers to a reported size limit of <70 kDa for a tat-conjugate peptide to readily cross cell membranes. This statement is now clarified and a citation provided.
- We have now gone through the legends to add information on animal age, and stated in the Results that FMRP(1-297) in slice experiments were all delivered through the electrode by either including it in the electrolyte, or actively exchanging the electrode solutions (ALA instruments). In case of infusion through the electrode, the effects of FMRP(1-297) can be detected at a stable level by 10-15 min after initiating the solution exchange process.

5. It is stated “We tail-vein injected 500 nM FMRP1-297-tat...”. What was the actual amount injected (also volume and length of infusion time)? Most of the results are from 30 min after the injection.

Further details on the amount and time of injecting FMRP(1-297)-tat are now expanded in the Methods section. The infusion time to inject FMRP(1-297)-tat is only 1-2 min, with time delays until animals are processed for histology or protein biochemistry identified in each Figure.

How does the level of expression compare at the 12 hour time measured, and at later times (e.g. after 3 days and 1-2 weeks)? In other words, what is the time frame of FMRP1-297-tat expression as determined by IF and WB?

This question was originally difficult to answer given that the effective concentration of FMRP(1-297)-tat proves to be so low as to present challenges for detection sensitivity of western blotting. By enriching the sample through immunoprecipitation with an HA antibody we were able to detect FMRP(1-297)-tat after injection in brain lysates to provide a relative comparison of band density over time (Fig. 5c). We also added a new figure comparing the relative intensities of immunofluorescence from 1-48 hr in cerebellum and cortex (Fig. 5a,b) by reacting all tissue simultaneously and imaging/processing with exactly the same parameters. Inasmuch as immunofluorescence intensity can be qualitatively compared these data do raise the interesting prospect that FMRP-tat intensity of label also increases over the initial 24 hr time and then has essentially dissipated by 48 hr.

6. How old and what sex were the animals that were injected via the tail vein?

This information is now provided in the Methods or within the specific section of Results or each Figure legend.

7. The motor activity data in Fig. 7 are quite interesting and the analysis is comprehensive. It is commendable that the authors also studied effects in wt mice and tested different doses. The results show a corrective effect on motor hyperactivity and velocity, but only at a single dose and only immediately after tail vein infusion of FMRP1-297-tat. As noted above, the manuscript would be improved by quantitating FMRP1-297-tat in the CNS at appropriate time points after infusion.

The reviewer is correct in that significant effects by FMRP1-297-tat on the OFT were immediate although

relatively short lasting in terms of average values of this behavioural measure. We note that some data sets showed very promising trends in reducing hyperactivity at 24 hr but this did not prove to be statistically significant when the final population was compiled. It was encouraging however that all three doses had no discernable behavioral effect on WT animals that have a full complement of FMRP, supporting the conclusion that the treatment option tested here is non-toxic, as concluded from data in **Figure 6** on cultured granule cells. It is also interesting that with the addition of a new **Figure 5** we establish that immunolabel for FMRP(1-297)-tat (HA identified) can be detected up to 48 hr after injection (**Fig. 5c**). In addition, the new measures of protein levels in Fmr1 KO mice show that FMRP(1-297)-tat can have relatively long-lasting effects on some aspects of protein translation even 24 hr after a single injection at 1.0 mg/kg (**Fig. 9**). In terms of quantifying FMRP(1-297)-tat, please see relative measures of this protein in brain lysates using a colP approach to enrich the samples in a new **Fig. 5**, and the response to Reviewer #1 under "FMRP associates with Cav3.1 and Kv4.3 channels"

8. In the discussion it is stated that:

"Another strategy is to use a tat-conjugate peptide, which we introduced by intravenous administration. An advantage to this is the ability to define the initial concentrations applied, avoiding the negative effects of overexpressing FMRP." This statement is somewhat misleading in that simply knowing how much is injected into the tail does not in itself avoid negative effects of over-expressing the protein. In fact, very little data were shown that addressed this point in terms of pan-CNS expression after peripheral administration.

Agreed. Given the space required for discussion of new findings this statement was simply removed.

9. Overall, an important observation in this study is that rapid reversal of synaptic plasticity and other abnormal parameters in fmr1 KO mice after infusion of FMRP1-297-tat. This is an important confirmation that the phenotype of FXS can be reversed during or after the developmental stage. The possible short term nature of the effects might limit the development of this approach in terms of a therapeutic. Are there any work-arounds that address this key issue that could be tried in the future and mentioned in the Discussion?

We added to the Discussion some considerations and references on how BBB transport and cell penetrance of FMRP-tat conjugates might be improved in the future by modifying the FMRP fragment or *tat* peptide. Our new findings on the duration of FMRP-tat effects on protein translation over at least 24 hr provide encouragement that many of the effects of this compound on circuit function long outlast the apparent shorter term effects on the specific behaviour of hyperactivity tested here.

Minor comments

FMRP1-297 is described as a "short fragment of FMRP" but cannot be described as such because it is about one half of the protein.

We removed the descriptor for FMRP(1-297) as being "short" from the text.

What is the "full-length FMRP-mKate construct"?

This refers to an mKate conjugate to the full length FMRP molecule (ie not FMRP(1-297)). To be more clear we changed the two references to this phrase to "FMRP-mKate construct" and explain this better in the text.

Reviewer #3 (Remarks to the Author):

The authors present an interesting approach to restoring functions affected in Fragile X syndrome. Injection of a blood-brain-barrier crossing peptide containing a short N-terminal fragment of FMRP restores specific neural circuit functions in a mouse model for Fragile X (Fmr1 KO mice), which rest on the complex forming capacity of this fragment with Cav3 and Kv4. The authors argue that this demonstration may provide the basis for future attempts to reduce FXS symptoms using peptide fragment injections. A link to behavioral output is also provided: peptide

delivery reduces hyperactivity, which is seen in these mice.

This paper is of interest, but of very limited scope. The question does arise to what degree this is a useful proof of principle demonstration. The interaction of FMRP with the Cav3-Kv4 complex is certainly important, but there is currently no evidence that interruption of this interaction contributes to any symptom in FXS (never mind that there is not even an established correlation between granule cell excitability and hyperactivity). Thus, the rescue effect, if limited to this N-terminal fragment and its functional roles, might be marginal.

Please see below and the final section of the Discussion for our interpretations of the relationships between mossy fiber plasticity and the hyperactivity measured here as a behaviour that is disrupted in *Fmr1* KO mice.

There are **two possible routes** the authors could go to make this study more impactful. First, to study effects on sensory processing in mice that result from FMRP deletion from cerebellar granule cells. This approach would involve the generation of granule cell specific FMRP deletion, and demonstration of sensory rescue in behaving mice.

Second, and alternatively, the authors might want to show that a similar effect can be observed on protein translation with a focus on peptides mimicking protein fragments involved in CYFIP1 and eIF4E interactions through which FMRP controls translation. In the context of this paper –that makes the point of the general usefulness of peptide injection strategies – it would be sufficient to test expression levels of specific marker proteins, without further behavioral testing.

We thank the reviewer for this suggestion, and have now included a new **Figure 9** where we tested the effects of FMRP(1-297)-tat injection on the levels of three proteins in both cerebellar and brain lysates 24 hr later. These results are very encouraging in showing that FMRP(1-297)-tat can reduce baseline differences in each of CaMKII , APP and PSD-95 to near WT levels that is detectable 24 hr later. Moreover, these findings help demonstrate that the actions of FMRP(1-297)-tat are widespread, with influence on molecular aspects of cell function that can be detected over an even longer timeframe than the average data on behavioural effects tested here. These data would support the prediction that FMRP(1-297)-tat will influence activity and circuit function in numerous regions beyond cerebellum.

It is true that both suggestions go a bit beyond the scope of the manuscript as submitted. That is, because as it is, the study is appealing from an experimental perspective, but will be of little consequence. The relation of FMRP function in excitability control toward FXS phenotypes is too speculative, there is not a single behavioral phenotype that has been linked to it. This also holds true for the observed hyperactivity as there is no evidence that the injected peptide acts through its effects on the Cav3-Kv4 complex in granule cells. It also remains possible that the N-terminal fragment interacts with other, unrelated protein complexes.

The behaviour of hyperactivity was tested in these trials due to an established link between hyperactivity in FXS patients that correlates to their specific level of FMRP expression and hypoplasia of the cerebellar vermis, especially in caudal lobules (Reiss et al., 1995; Gothelf et al., 2008; Hampson and Blatt, 2015). We would agree that the effects of FMRP-tat on cellular activity and synaptic plasticity in granule cells has no established direct link to hyperactivity. This is now more clearly distinguished in the Discussion, along with our intention of using the mossy fiber-granule cell synapse essentially as a test case for which we understand an ionic basis for Cav3 and Kv4 channels in contributing to LTP at this synapse. In this respect mossy fiber LTP and Cav3/Kv4 function serves well to verify concentration dependent effects after tail vein injections of FMRP(1-297)-tat on cell activity. The results obtained in new tests to respond to reviewers' requests to consider its distribution over time in other brain regions and protein translation alleviate any concern that we are implying that all the effects of this protein are centered on plasticity in cerebellum. Rather, we now document the presence and persistence of HA immunolabel up to 48 hr in both cerebellar and cortical regions, and the ability for FMRP(1-297)-tat to remove standing differences in the levels of three key proteins that have been previously studied in FXS. By extending those tests to both brain and

cerebellar lysates we provide strong evidence and state categorically that any behavioural effects observed must undoubtedly reflect its action on widespread brain regions, rather than cerebellum alone.

References:

- Anderson D, Engbers JD, Heath NC, Bartoletti TM, Mehaffey WH, Zamponi GW, Turner RW (2013) The Cav3-Kv4 complex acts as a calcium sensor to maintain inhibitory charge transfer during extracellular calcium fluctuations. *J Neurosci* 33:7811–7824.
- Anderson D, Mehaffey WH, Iftinca M, Rehak R, Engbers JD, Hameed S, Zamponi GW, Turner RW (2010) Regulation of neuronal activity by Cav3-Kv4 channel signaling complexes. *Nat Neurosci* 13:333–337.
- Gothelf D, Furfaro JA, Hoeft F, Eckert MA, Hall SS, O’Hara R, Erba HW, Ringel J, Hayashi KM, Patnaik S, Golianu B, Kraemer HC, Thompson PM, Piven J, Reiss AL (2008) Neuroanatomy of fragile X syndrome is associated with aberrant behavior and the fragile X mental retardation protein (FMRP). *Ann Neurol* 63:40–51.
- Hampson DR, Blatt GJ (2015) Autism spectrum disorders and neuropathology of the cerebellum. *Front Neurosci* 9:420.
- Heath NC, Rizwan AP, Engbers JD, Anderson D, Zamponi GW, Turner RW (2014) The expression pattern of a Cav3-Kv4 complex differentially regulates spike output in cerebellar granule cells. *J Neurosci* 34:8800–8812.
- Reiss AL, Abrams MT, Greenlaw R, Freund L, Denckla MB (1995) Neurodevelopmental effects of the FMR-1 full mutation in humans. *Nat Med* 1:159–167.
- Rizwan AP, Zhan X, Zamponi GW, Turner RW (2016) Long-term potentiation at the mossy fiber-granule cell relay invokes postsynaptic second-messenger regulation of Kv4 channels. *J Neurosci* 36:11196–11207.

Reviewers' Comments:

Reviewer #2:

Remarks to the Author:

The paper is much improved with more details and greater clarity. However, there are still several issues to be rectified (points 1-3 below), plus 1 recommendation (point 4).

1. In Fig. 4e-f there is a key control missing: Fmr1 KO not injected with tat and labeled with anti-HA. The control shown in 4e (anti-MAP+HA) is not useful for this purpose which is to show a clear demonstration of the background in an uninjected Fmr1 mouse with anti-HA only. The authors state that within 30 min of injection, granule cells are labeled. I cannot tell if this is true w/o this key control.

2. In the time study shown in Fig. 5 the authors should clarify in the legend, if correct, that all panels shown including vehicle controls are labeled with anti-HA.

In Fig. 5a the authors again state that granule neurons are labeled but I can't really see this except for maybe at the 12 hr where there is a large red sphere in the GC layer which may or may not be a granule cell. The authors should consider backing off on the claim of granule cell expression unless these issues can be addressed.

3. The results of the WB shown in Fig. 5c are not very convincing as the bands are very faint across all time points. The authors did not - but should - indicate in the legend what exactly is represented in the first lane containing the control. It is difficult to see much difference across the lanes - possible non-specific? Also, unlike what is stated in the results text, this result with the WB does not seem to coincide with the panels in 5a,b.

In the absence of more convincing WB detection, the authors could consider removing the WB result in 5c.

4. In the abstract the authors say "... reduces hyperactivity in adult Fmr1 KO mice within 1 hour..." and in the Discussion they state "...and reduces hyperactivity in Fmr1 KO animals." To convey full transparency, more accurate wording would be "... transiently reduces hyperactivity in adult Fmr1 KO mice..." and "...and reduces hyperactivity in Fmr1 KO animals at 1 hour post-FMRP-tat injection, but not at 24 or 48 hours post-injection."

5. Minor suggestion: The revised Discussion has become quite verbose. This reviewer believes that deleting (preferred) or moving the last paragraph of the Discussion (section titled FMRP regulation of protein translation) and ending the paper with the sentence "Thus, we fully expect that the effects of this FMRP fragment on OFT as a representative behaviour reflects actions on circuit output across widespread brain regions" would be better. However, this is a judgement call.

Reviewer #3:

Remarks to the Author:

The authors have been responsive to my previous comments. In particular, they have tested the effects of the FMRP(1-297)-tat rescue peptide on protein translation (which is regulated by FMRP). This is all very good.

At the same time, I remain to feel uneasy about the inability to connect dots. The authors are able to show that specific alterations in KO mice are restored to normal with the rescue peptide.

However, it remains true that it is entirely unclear how (and if) these specific alterations relate to Fragile X syndrome in the first place. Granule cell excitability is not generally related to hyperexcitability in the Fragile X brain. That remains a problem. Possibly, the new finding on restoring some alterations in protein translation is more telling.

We thank the reviewers for their constructive comments and provide all the requested changes to the text, modified Figs. 4 and 5, and reduced the Discussion in length for clarity. All responses are provided below.

Reviewers' comments:

Reviewer #2 (Remarks to the Author):

The paper is much improved with more details and greater clarity. However, there are still several issues to be rectified (points 1-3 below), plus 1 recommendation (point 4).

1. In Fig. 4e-f there is a key control missing: Fmr1 KO not injected with tat and labeled with anti-HA. The control shown in 4e (anti-MAP+HA) is not useful for this purpose which is to show a clear demonstration of the background in an uninjected Fmr1 mouse with anti-HA only. The authors state that within 30 min of injection, granule cells are labeled. I cannot tell if this is true w/o this key control.

We have now revised the figure to show both the dual labeled MAP-2 & HA and the HA labeled section alone from an animal not injected with HA-FMRP-tat in Fig. 4e. We believe that this more clearly shows the difference encountered between animals injected with vehicle alone compared to those injected with HA-FMRP-tat (Fig. 4f, g).

2. In the time study shown in Fig. 5 the authors should clarify in the legend, if correct, that all panels shown including vehicle controls are labeled with anti-HA.

We have now added this statement to the Figure Legend to clarify that all tissue sections shown in Fig 5a, b were treated with the anti-HA antibody.

In Fig. 5a the authors again state that granule neurons are labeled but I can't really see this except for maybe at the 12 hr where there is a large red sphere in the GC layer which may or may not be a granule cell. The authors should consider backing off on the claim of granule cell expression unless these issues can be addressed.

We can state that granule cell labeling was often present, but was more obvious in Purkinje cells. The reviewer is correct that in the time series reacted in Fig 5a labeling in granule cells was not as clear as Purkinje cells, but granule cells can be seen in Fig. 4f 30 min after tail vein injection of FMRP(1-297), and in Supplemental Fig. 6 where injection of only 100 nM FMRP(1-297)-tat was sufficient to label the granule cells after 2 hr. We now clarified the description of Results shown in Fig 5 to ensure that it accurately reflects what this particular set of data shows.

3. The results of the WB shown in Fig. 5c are not very convincing as the bands are very faint across all time points. The authors did not - but should - indicate in the legend what exactly is represented in the first lane containing the control. It is difficult to see much difference across the lanes – possible non-specific?

The control on the previous figure represented a Fmr1 KO mouse injected with only vehicle, which reported a negative label when immunoblotted with the N-terminal FMRP antibody. From here we can confirm the specificity of the band of FMRP, which is also shown by the positive control lane labeled for HA-FMRP(1-297).

Also, unlike what is stated in the results text, this result with the WB does not seem to coincide with the panels in 5a,b. In the absence of more convincing WB detection, the authors could consider removing the WB result in 5c.

We agree, and certainly found that detecting injected FMRP by WB was difficult. Of interest is that a separate set of tests we conducted in cultured granule cells showed that WB can only detect FMRP(1-297)-HA in cell lysates after it was applied directly to the cultured medium (ie no blood brain barrier) at concentrations above 10 nM and left for 1-6 hr. The difficulty in detecting this label in brain lysates after tail vein injection must then indicate that the effective dose of 1 mg/kg results in final concentrations in central neurons at levels lower than 10 nM. In support of this, another study recently showed that FMRP levels only need to increase by 5% to find a reversal of excitability in human iPSC cells (Graef et al. 2019). Therefore, we have taken the advice of the reviewer and removed the WB detection previously shown in Fig 5c.

Graef, J. D. *et al.* Partial FMRP expression is sufficient to normalize neuronal hyperactivity in Fragile X neurons. *Eur. J. Neurosci.* (2019) doi:10.1111/ejn.14660

4. In the abstract the authors say “.... reduces hyperactivity in adult Fmr1 KO mice within 1 hour...” and in the Discussion they state “...and reduces hyperactivity in Fmr1 KO animals.” To convey full transparency, more accurate wording would be “.... transiently reduces hyperactivity in adult Fmr1 KO mice...” and “...and reduces hyperactivity in Fmr1 KO animals at 1 hour post-FMRP-tat injection, but not at 24 or 48 hours post-injection.”

This alternate wording in the Abstract and Discussion has now been inserted.

5. Minor suggestion: The revised Discussion has become quite verbose. This reviewer believes that deleting (preferred) or moving the last paragraph of the Discussion (section titled FMRP regulation of protein translation) and ending the paper with the sentence “Thus, we fully expect that the effects of this FMRP fragment on OFT as a representative behaviour reflects actions on circuit output across widespread brain regions” would be better. However, this is a judgement call.

We feel that the section on protein translation is important to keep given that Reviewer 3 had requested these experiments, but we moved this content farther up into the Discussion to end on the paragraph suggested here. We also went through the rest of the Discussion and found other locations to trim the length by a total of 330 words (~22% shorter).

Reviewer #3 (Remarks to the Author):

The authors have been responsive to my previous comments. In particular, they have tested the effects of the FMRP(1-297)-tat rescue peptide on protein translation (which is regulated by FMRP). This is all very good.

At the same time, I remain to feel uneasy about the inability to connect dots. The authors are able to show that specific alterations in KO mice are restored to normal with the rescue peptide. However, it remains true that it is entirely unclear how (and if) these specific alterations relate to Fragile X syndrome in the first place. Granule cell excitability is not generally related to hyperexcitability in the Fragile X brain. That remains a problem. Possibly, the new finding on restoring some alterations in protein translation is more telling.

Our findings do not actually speak towards granule cell excitability, as this did not differ between FMRP KO and WT animals. Rather, the ability for FMRP to enable LTP of mossy fiber input provides another example where FMRP can control plasticity of signal processing, and in this case a large bank of sensory input. We agree that there is no a priori reason for this specific form of plasticity to be any more relevant to hyperactivity in FMRP KO mice than the actions of FMRP on LTP or LTD in many other

brain regions. We try to make this as clear as we can in the Discussion by making the statement “It is important to clarify that we can not conclude that the rescue of mossy fiber plasticity by FMRP(1-297)-*tat* is directly linked to hyperactive behaviours in the OFT.” And then clarify that once the work moved into the whole animal and tail vein injections, this form of plasticity served as the base test of effects by injected FMRP(1-297)-*tat* at the cellular and synaptic level.

Reviewers' Comments:

Reviewer #2:

Remarks to the Author:

n/a

Reviewer #3:

Remarks to the Author:

The authors have addressed technical issues raised by this and other reviewers. The revision is appropriate from this point of view. My more conceptual concerns remain.

REVIEWERS' COMMENTS:

Reviewer #2 (Remarks to the Author):

n/a

Reviewer #3 (Remarks to the Author):

The authors have addressed technical issues raised by this and other reviewers. The revision is appropriate from this point of view. My more conceptual concerns remain.

We believe the reviewer is referring to interpretations related to hyperactivity in Fragile X Syndrome and its specific relation to the cell and synaptic data sets presented here. As indicated in the Results and Discussion, we now only refer to relative levels of activity in the Open Field Test (as compared to hyperactivity) and mention in the Discussion that this activity can also be interpreted in terms of anxiety. However, no specific interpretation is applied here pending future studies that address anxiety per se. We also state that we used activity levels in the Open Field Test as a readily quantifiable behaviour that is known to differ between Fmr1 KO and wild type mice, and that this very likely reflects circuit output over a wide range of brain regions as compared to cerebellum alone. With the addition of protein level tests requested by the reviewer we feel that the marked reduction of differences between wt and KO animals 24 hr after FMRP(1-297)-tat injection supports the proposal that the actions of this compound on activity levels in the OFT can be expected to derive from widespread regions yet to be identified.